# Crucial role for T cell-intrinsic IL-18R-MyD88 signaling in cognate immune response to intracellular parasite infection

Ana-Carolina Oliveira[1†‡], João Francisco Gomes-Neto[1†], Carlos-Henrique Dantas Barbosa[1], Alessandra Granato[1], Bernardo S Reis[2], Bruno Maia Santos[1], Rita Fucs[3], Fábio B Canto[1], Helder I Nakaya[4,5,6], Alberto Nóbrega[1], Maria Bellio[1,5]*

[1]Instituto de Microbiologia Paulo de Góes, Universidade Federal do Rio de Janeiro, Rio de Janeiro, Brazil; [2]The Rockefeller University, New York, United States; [3]Instituto de Biologia, Universidade Federal Fluminense, Niterói, Brazil; [4]Faculdade de Ciências Farmacêuticas, Universidade de São Paulo, São Paulo, Brazil; [5]Instituto Nacional de Ciência e Tecnologia de Vacinas, CNPq-MCT, Belo Horizonte, Brazil; [6]Department of Pathology, Emory University School of Medicine, Atlanta, United States

*For correspondence:
mariabellioufrj@gmail.com

[†]These authors contributed equally to this work

Present address: [‡]Instituto de Biofísica Carlos Chagas Filho, Universidade Federal do Rio de Janeiro, Rio de Janeiro, Brazil

Competing interests: The authors declare that no competing interests exist.

**Abstract** MyD88 is the main adaptor molecule for TLR and IL-1R family members. Here, we demonstrated that T-cell intrinsic MyD88 signaling is required for proliferation, protection from apoptosis and expression of activation/memory genes during infection with the intracellular parasite *Trypanosoma cruzi*, as evidenced by transcriptome and cytometry analyses in mixed bone-marrow (BM) chimeras. The lack of direct IL-18R signaling in T cells, but not of IL-1R, phenocopied the absence of the MyD88 pathway, indicating that IL-18R is a critical MyD88-upstream pathway involved in the establishment of the Th1 response against an *in vivo* infection, a presently controvert subject. Accordingly, $Il18r1^{-/-}$ mice display lower levels of Th1 cells and are highly susceptible to infection, but can be rescued from mortality by the adoptive transfer of WT CD4[+] T cells. Our findings establish the T-cell intrinsic IL-18R/MyD88 pathway as a crucial element for induction of cognate Th1 responses against an important human pathogen.
DOI: https://doi.org/10.7554/eLife.30883.001

## Introduction

*Myd88*-deficient animals are highly susceptible to infection by multiple pathogens, including *T. cruzi*, and exhibit diminished Th1 differentiation, which has been attributed to defective IL-12 production by APCs, as a consequence of poor TLR signaling (*Campos et al., 2004*; *Fremond et al., 2004*; *Scanga et al., 2002*; *Seki et al., 2002*; *Muraille et al., 2003*). Few studies to date have directly addressed the relevance of T cell-intrinsic MyD88 signaling pathways for the establishment of in vivo cognate Th1 responses in the context of infection (*Frazer et al., 2013*; *LaRosa et al., 2008*; *Raetz et al., 2013*; *Zhou et al., 2009*). Although these studies reported that the absence of T-cell intrinsic MyD88 signaling severely impact the immune response, the Toll/IL-1R homologous region (TIR) domain-containing receptor upstream of MyD88 acting on CD4[+] T cells was either not investigated or not identified and, therefore, remains speculative. Thus, presently, no consensus exists about the relative contribution of different receptors upstream MyD88 necessary for sustaining a robust Th1 response and contributing to CD4[+] T cell memory formation in a model of infection.

Cytokines of the IL-1 family contribute for the reinforcement and/or stabilization of CD4$^+$ T cell lineage commitment into each of the main Th phenotypes: Th17, Th1 and Th2 (*Acosta-Rodriguez et al., 2007*; *Chung et al., 2009*; *Guo et al., 2009*). While the essential contribution of direct IL-1R signaling for the differentiation of Th17 cells has been documented in the EAE mouse model (*Chung et al., 2009*), the direct effect of IL-1 or IL-33 on the expansion of Th1 cells remains a more controversial issue (*Ben-Sasson et al., 2009*; *Schenten et al., 2014*; *Villarreal and Weiner, 2014*). IL-18 was initially shown to synergize with IL-12 for IFN-γ production by Th1 cells *in vitro* (*Robinson et al., 1997*), but its essential role in promoting Th1 responses to infection was not always confirmed in the context of *in vivo* infection (*Haring and Harty, 2009*; *Monteforte et al., 2000*). Moreover, although in other circumstances *Il18$^{-/-}$* mice show a diminished Th1 response (*Takeda et al., 1998*), this phenotype cannot be uniquely ascribed to the lack of response of T cells to IL-18, as IL-18 also potentiates the secretion of IFN-γ by other cells, like NK cells (*Takeda et al., 1998*), which could in turn impact on Th1 response. In fact, NK-derived IFN-γ has a profound influence on Th1 responses (*Scharton and Scott, 1993*). Therefore, the full significance of T-cell intrinsic IL-1R and IL-18R signaling for Th1 responses to infection *in vivo* is still an important issue that needs further clarification.

To investigate the role of T-cell intrinsic MyD88 signaling on Th1 differentiation *in vivo*, here we have infected mixed bone marrow (BM) chimeric mice with the intracellular protozoan *T. cruzi*, the etiologic agent of Chagas' disease (*American trypanosomiasis*), a neglected emerging disease, which is endemic in Latin America. Several TLRs are involved in the protection against *T. cruzi* and *Myd88$^{-/-}$* mice are highly susceptible to infection, displaying low levels of IFN-γ$^+$CD4$^+$ T cells (*Bafica et al., 2006*; *Caetano et al., 2011*; *Campos et al., 2004*; *Oliveira et al., 2004, 2010*; *Rodrigues et al., 2012*). Although the absence of TLR signaling in APCs of *Myd88$^{-/-}$* mice may lead to their deficient activation and may explain a limited Th1 polarization response, these former results do not exclude the possibility that the absence of CD4$^+$ T cell-intrinsic MyD88 signaling through IL-1R family members could also be an important factor for the deficient levels of Th1 cells in *Myd88$^{-/-}$* mice. Here, we tested this hypothesis by comparing WT and *Myd88-*, *Il1r1-* or *Il18r1-* deficient T cells in infected mixed BM chimeras. Besides comparing T cell numbers, BrdU incorporation and the expression of IFN-γ, CCR5 and CD44 by flow cytometry, we also analyzed the transcriptional profile of WT and *Myd88-*deficient CD4$^+$ T cells, sorted from infected mixed BM chimeric mice. Our results revealed, for the first time, the critical role of T cell-intrinsic IL-18R/MyD88 signaling for mounting a robust Th1 cognate response, as a consequence of proliferation, protection from apoptosis and expression of activation/memory genes during infection with an intracellular pathogen. Furthermore, we demonstrated the high susceptibility of *Il18r1$^{-/-}$* mice to infection with *T. cruzi*, which could be rescued by the adoptive transfer of WT CD4$^+$ T cells. In summary, the present study unambiguously demonstrates the crucial role of T cell-intrinsic IL-18R/MyD88 signaling for a robust Th1 cognate response against an important human pathogen and discloses the mechanistic framework underlying it.

## Results

### *Myd88$^{-/-}$* Th1 cells attain lower frequencies in infected mixed BM chimeras

We first generated BM chimeras in which WT mice were irradiated and reconstituted with either WT or *Myd88$^{-/-}$* BM cells. Infected *Myd88$^{-/-}$*→WT mice die earlier than WT→WT chimeras, displaying mortality kinetics similar to non-chimeric *Myd88$^{-/-}$* and WT mice, respectively (*Figure 1A and B*). This result shows that chimeric mice lacking MyD88 expression exclusively in cells of hematopoietic origin present a higher level of susceptibility to infection, similar to *Myd88$^{-/-}$* mice. Next, we generated mixed BM chimeras. For this, irradiated WT B6 x B6.SJL F1 (CD45.1$^+$CD45.2$^+$) mice were reconstituted with a 1:1 mix of WT (CD45.1$^+$) and *Myd88$^{-/-}$* (CD45.2$^+$) BM cells (*Figure 1—figure supplement 1A*). Six to 8 weeks after reconstitution, non-infected mixed chimeric mice show equivalent frequencies of WT and *Myd88$^{-/-}$*CD8$^+$ and CD4$^+$ T cells (*Figure 1C–E*). Including residual WT recipient cells (CD45.1$^+$CD45.2$^+$), the mixed BM chimeras have more than 50% of WT DC in their spleen (*Figure 1—figure supplement 1B*) able to be fully activated by TLR pathways during infection with *T. cruzi*. At day 14 pi, we observed a preferential expansion of WT CD4$^+$ T cells at the

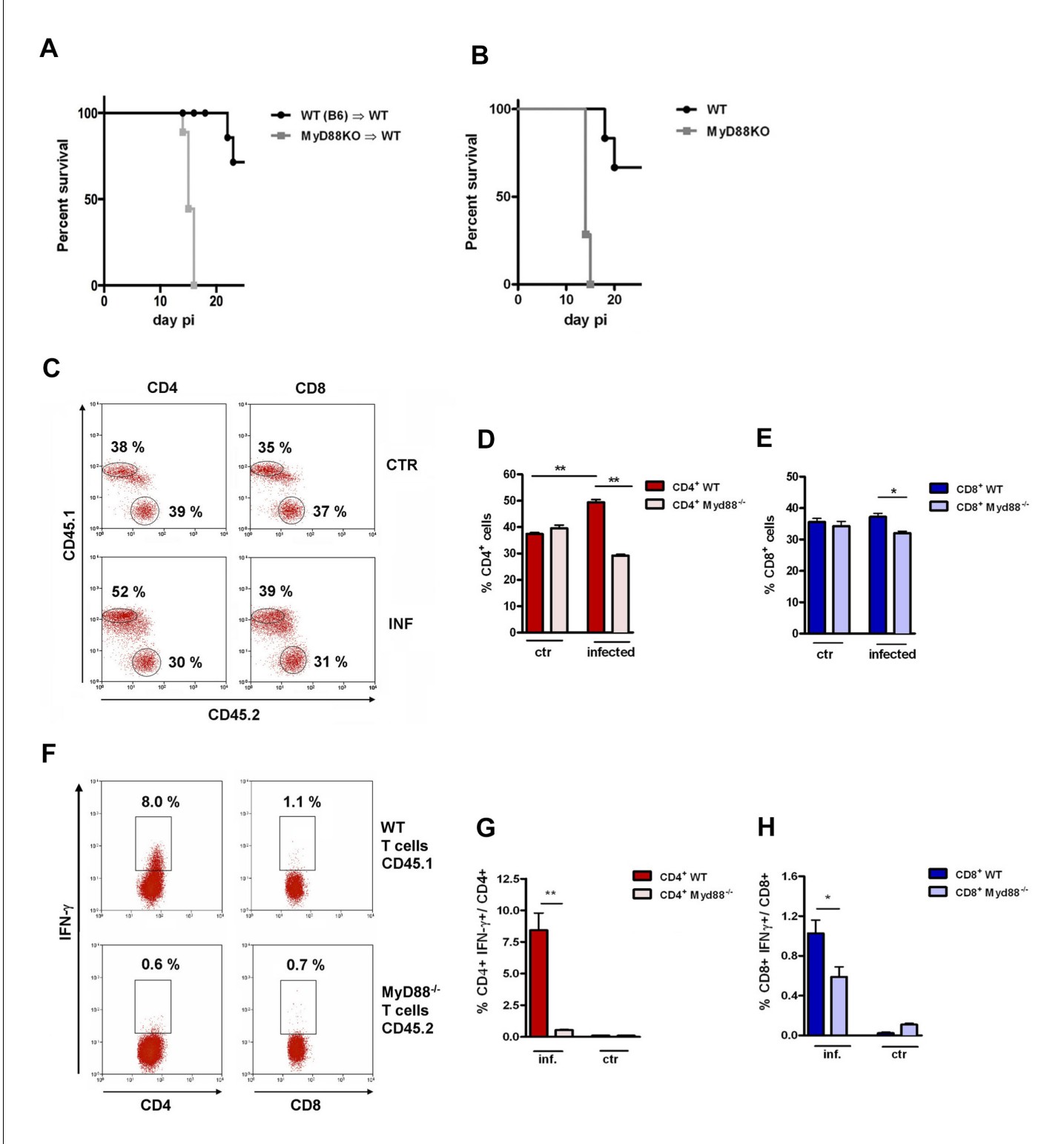

**Figure 1.** Lower expansion of IFN-γ+CD4+ *Myd88*-/- cells in infected mixed BM chimeras. (A, B) Survival curves of mice infected ip with 2 × 10³ blood trypomastigotes of the Y strain. (A) *Myd88*-/- (CD45.2+)→WT (B6 x B6.SJL F1, CD45.1+CD45.2+) and WT (B6.SJL, CD45.1+)→WT (B6 x B6.SJL F1, CD45.1+CD45.2+) chimeric mice 8 weeks after reconstitution and (B) WT (B6) and *Myd88*-/- mice. Survival curves are statistically different (p<0.05). All surviving mice in (A) were euthanized on day 25 pi (n = 6 to 9 per group). (C) Representative dot plots showing frequencies of CD45.1+ (WT) and CD45.2+ (*Myd88*-/-) among gated CD4+ or CD8+ spleen T cells from control non-infected or infected mixed BM chimeras. (D, E) Mean frequencies of

*Figure 1 continued on next page*

*Figure 1 continued*

data shown in (C). (F) Representative dot plots showing frequencies of IFN-γ⁺CD45.1⁺ and IFN-γ⁺CD45.2⁺ among gated CD4⁺ or CD8⁺ T cells. (G, H) Mean frequencies of data shown in (F). Mice (n = 4 per group) from control or infected mixed BM chimeric mice were Individually analyzed on day 14 pi. Error bars = SEM, *p≤0.05; **p≤0.01 (two-tailed Student $t$ test). Data are representative of 4 independent experiments. Experimental design, frequency of CD45.1⁺ (B6.SJL, WT) and CD45.2⁺ (*Myd88⁻/⁻*) gated on CD11c^high spleen cells, as well as the kinetics of IFN-γ⁺CD4⁺ T cell response are shown in *Figure 1—figure supplement 1*.

DOI: https://doi.org/10.7554/eLife.30883.002

The following figure supplement is available for figure 1:

**Figure supplement 1.** Mixed bone-marrow chimeras.

DOI: https://doi.org/10.7554/eLife.30883.003

expense of *Myd88⁻/⁻*CD4⁺ T cells and, to a lower extent, of WT CD8⁺ T cells, at the expense of *Myd88⁻/⁻*CD8⁺ T cells (*Figure 1C–E*). This scenario was even more dramatic when the frequencies of IFN-γ⁺ cells among WT and *Myd88⁻/⁻*CD4⁺ T cells were analyzed. In order to determine the frequency of Ag-specific IFN-γ-producing CD8⁺T cells, total splenocytes from infected mixed BM chimeras were cultured in the presence of the K^b-restricted TSKB20 peptide, an immunodominant CD8 epitope (*Oliveira et al., 2010*). Infected APCs, present in the spleen of infected mice, are able to induce Ag-specific stimulation of CD4⁺ T cells *in vitro,* without the need of adding extra *T. cruzi*-derived Ag into the cultures (*Oliveira et al., 2010*). As shown in *Figure 1F and G*, the frequency of IFN-γ⁺CD4⁺ T cells of *Myd88⁻/⁻* origin was severely diminished compared to the frequency of WT IFN-γ⁺CD4⁺ T cells. The levels of *Myd88⁻/⁻*IFN-γ⁺CD8⁺ T cells were also diminished compared to WT IFN-γ⁺CD8⁺ T cells, but to a lower extent (*Figure 1F and H*). In mixed BM chimeras, similar to what was found in WT B6 mice (*Oliveira et al., 2010*), CD4⁺IFN-γ⁺ T cells are first detected around day 10 pi, their numbers attain a maximum around day 14 pi and then begin to decline (*Figure 1—figure supplement 1C*). Therefore, kinetic differences cannot explain the observed disparity in the percentages of IFN-γ⁺ cells between WT and *Myd88⁻/⁻* CD4⁺ T cells.

We next confirmed the lower expansion of *Myd88⁻/⁻*IFN-γ⁺CD4⁺ T lymphocytes in non-irradiated *Rag2⁻/⁻* recipient mice reconstituted with a 1:1 mix of WT and *Myd88⁻/⁻* splenocytes. In *Rag2⁻/⁻* recipient mice, 100% of host APCs express MyD88 and, again, we found that percentages and absolute numbers of CD4⁺IFN-γ⁺ T cells were much lower among *Myd88⁻/⁻*, than among WT cells (*Figure 2*). The same was true for IFN-γ-producing CD8⁺ T cells (*Figure 2—figure supplement 1*). Together, these results clearly demonstrate that T cell-intrinsic MyD88 signaling is required for the expansion of Th1 cells, even when the totality of APCs in recipient mice expresses MyD88 and, therefore, are fully competent to provide co-stimulatory signals and cytokines.

## WT and *Myd88⁻/⁻* CD4⁺ T cells of infected mixed BM chimeras display different gene-expression programs

We then compared gene expression between WT and *Myd88⁻/⁻*CD4⁺ T cells sorted from the spleen of infected mixed BM chimeras (*Figure 3A*). Several Th1 related genes as *Ifng, Ccr5, Ccl5, Ccl4 and Ccl3* were upregulated in WT CD4⁺ T cells compared to *Myd88⁻/⁻*CD4⁺ T cells (*Figure 3B*). Table 1 (*Figure 3—source data 1*) lists genes upregulated and downregulated in CD4⁺ T cells from *Myd88⁻/⁻* vs WT (B6) origin. Gene expression differences between WT and *Myd88⁻/⁻* CD4⁺ T cells were then used in Gene Set Enrichment Analysis (GSEA) pre-ranking analysis (*Figure 3C and D*). Interestingly, sets of genes related to central and effector memory CD4⁺ T cells (CM and EM, respectively) (*Abbas et al., 2009*) were upregulated in WT cells, while genes expressed in naïve cells were relatively upregulated in *Myd88⁻/⁻*CD4⁺ T cells (*Figure 3C*). Moreover, sets of genes related to anti-apoptotic pathways, as well as to cell cycle and mitosis were upregulated in WT CD4⁺ T cells (*Figure 3D*), in accordance with the higher expansion of WT cells observed in mixed BM chimeras (*Figure 1*) and with previous studies reporting that *Myd88⁻/⁻* T cells have increased propensity for apoptosis (*Frazer et al., 2013*; *Tomita et al., 2008*). Interactions among the genes were assessed and visualized by Ingenuity Pathway Analysis (IPA) database, indicating a network of naïve CD4⁺ T cell genes, which are highly expressed in *Myd88⁻/⁻*CD4⁺ T cells (*Figure 3—figure supplement 1A*) and a network of EM genes highly expressed in WT CD4⁺ T cells (*Figure 3—figure supplement 1B*). Therefore, the analyses of RNA microarray data are in conformity with our cytometry results and indicate that CD4⁺ T cell-intrinsic MyD88 signaling is necessary for sustaining a more robust Th1

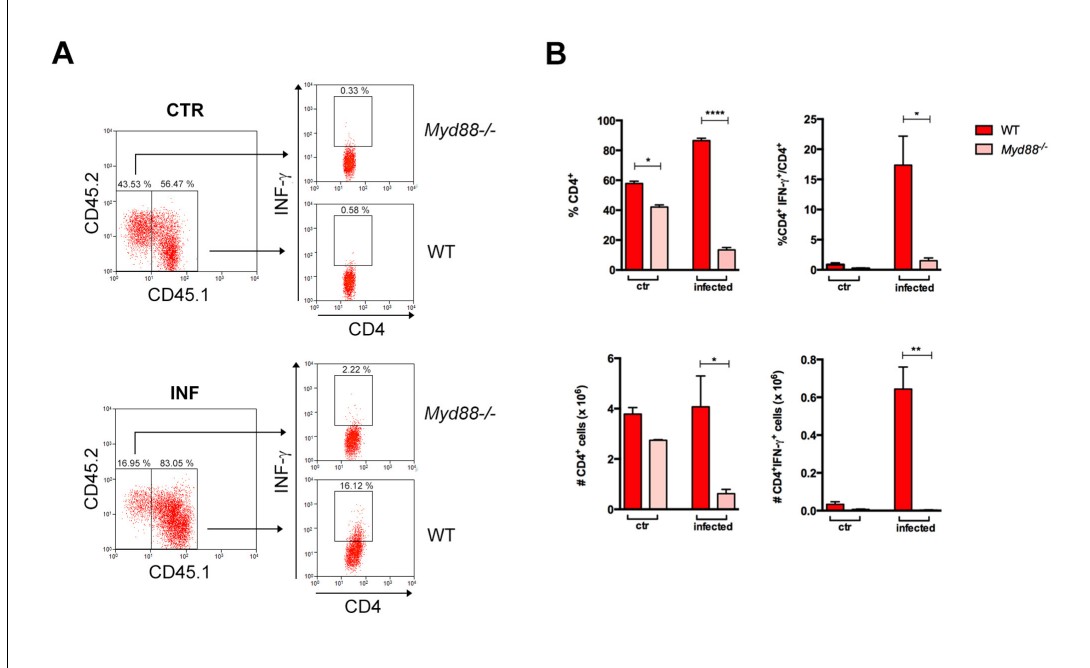

**Figure 2.** Lower expansion of IFN-γ⁺CD4⁺ *Myd88* ᵀ cells in infected *Rag2⁻/⁻* mice reconstituted with WT and *Myd88⁻/⁻* splenocytes. (**A**) Gate strategy and dot plot analysis of IFN-γ⁺CD4⁺ T cells among CD4⁺CD45.2⁺ (*Myd88⁻/⁻*) or among CD4⁺CD45.1⁺ (B6) plus CD4⁺CD45.2⁺CD45.1⁺(B6xB6.SJL F1) (WT) T cells in *Rag2⁻/⁻* mice, reconstituted as described in Material and Methods section. (**B**) Mean frequencies and absolute cell numbers of *Myd88⁻/⁻* (rose) and WT (red) total CD4⁺ or IFN-γ⁺CD4⁺ T cells gated on CD4⁺ T cells from reconstituted naïve or infected *Rag2⁻/⁻* mice, individually analyzed on day 14 pi (n = 3 to 4). Error bars = SEM, *p≤0.05; **p≤0.01; ****p≤0.000 (two-tailed Student *t*-test). Data are representative of 2 independent experiments. Results obtained for the CD8⁺ T cell subpopulation are shown in *Figure 2—figure supplement 1*.

DOI: https://doi.org/10.7554/eLife.30883.004

The following figure supplement is available for figure 2:

**Figure supplement 1.** Lower expansion of IFN-γ⁺CD8⁺ *Myd88⁻/⁻* cells in infected *Rag2⁻/⁻* mice reconstituted with WT and *Myd88⁻/⁻* splenocytes.
DOI: https://doi.org/10.7554/eLife.30883.005

differentiation program, proliferation and resistance to apoptosis, resulting in selective expansion of WT IFN-γ⁺CD4⁺ T cells in response to infection with *T. cruzi*.

## Lack of T cell-intrinsic IL-18R or MyD88 signaling leads to lower frequencies and numbers of Th1 cells but does not affect CD8⁺CTLs

Since MyD88 is an essential adaptor molecule not only downstream of most TLRs, but also of IL-1 and IL-18 pathways, we investigated whether the lack of IL-1R or IL-18R would by itself reproduce the effects observed in *Myd88⁻/⁻* T cells. First, we performed a kinetic study of the presence of IL-1β and IL-18 in the serum of infected B6 mice. As shown in *Figure 4—figure supplement 1A*, a higher level of IL-1β was detected on day 13 pi, although an early peak of this cytokine was also detected at 12 hr after infection. On the other hand, we could only detect IL-18 at a later time point of infection, at day 13 pi (*Figure 4—figure supplement 1B*). Note that peaks of IL-1β and IL-18 at day 13 pi were paralleled by increased percentages of Th1 cells in the spleen of infected mice (*Figure 4—figure supplement 1C*). We then analyzed the expression of IL-1R1 and IL-18R1 receptors on splenic T cells from infected B6 mice (at day 14 pi) and from non-infected controls. As shown in *Figure 4A*, at this time point of infection, around 50% of CD44ʰⁱCD4⁺ T cells express the master gene of Th1 cells, T-bet, and the majority of T-bet⁺CD44ʰⁱCD4⁺ T cells expresses IL-18R1 but not IL-1R1 (*Figure 4B*). In fact, no upregulation of IL-1R1 was observed on CD44ʰⁱCD4⁺ T splenocytes from infected mice (*Figure 4B*) and a higher percentage of T-bet⁺CD44ʰⁱCD4⁺ T cells expresses IL-1R1 in non-infected controls than in infected mice (*Figure 4C*). To further dissect this issue, we stained splenic T cells from infected B6 mice with anti-CD44 and anti-CD62L. As shown in *Figure 4D*, IL-

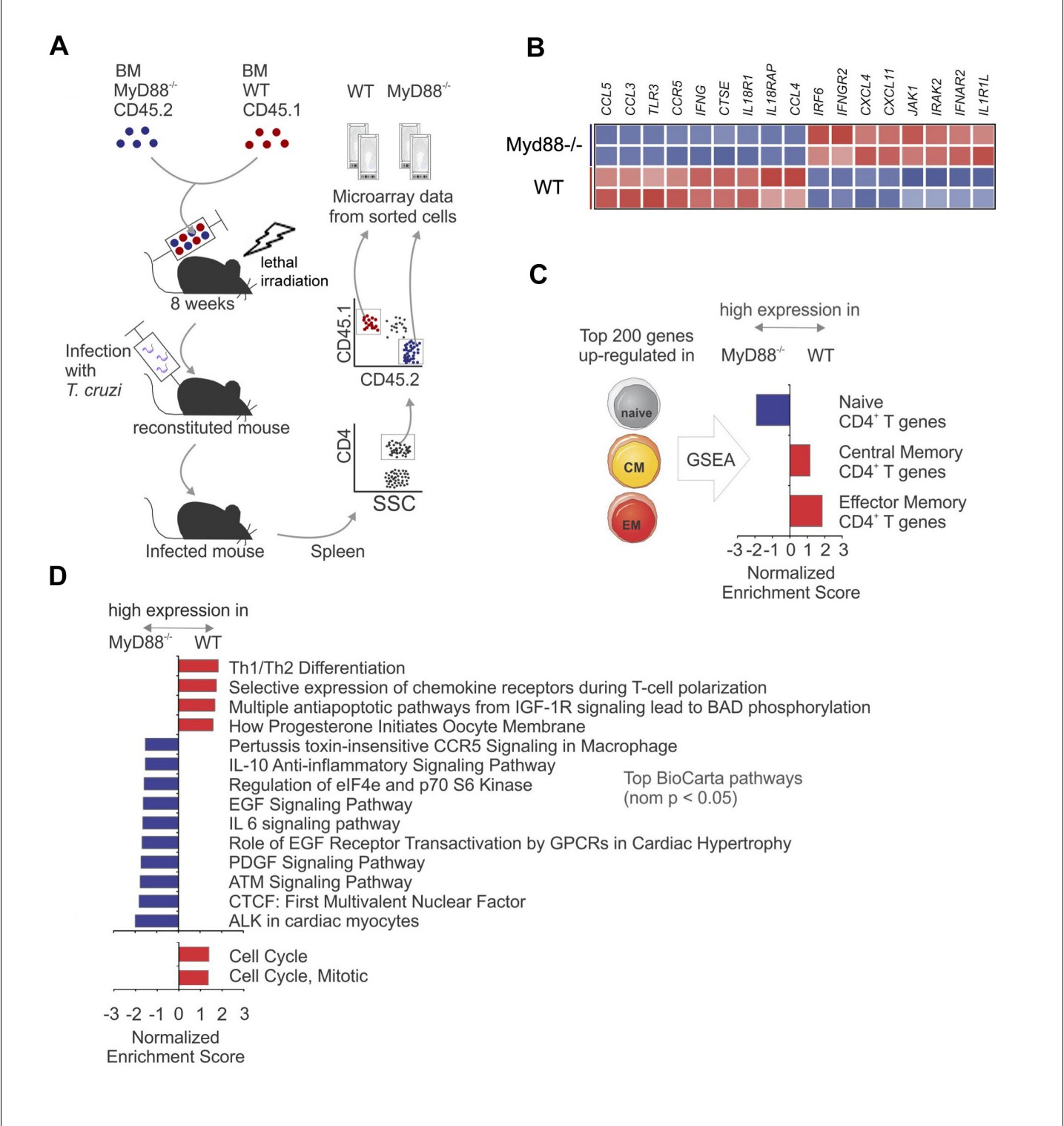

**Figure 3.** Gene expression profiles of WT and *Myd88−/−* CD4+T cells sorted from mixed BM chimeras infected with *T. cruzi.* (**A**) Experimental design: CD4+CD45.1+ cells (WT) and CD4+CD45.2+ cells (*Myd88−/−*) were sorted (99% of purity) from the pool of splenocytes obtained from infected mixed BM chimeric mice at day 14 pi (n = 4). Two individually sorted samples of each cell population (WT or *Myd88−/−*) were assayed independently for gene expression profile. (**B**) Heat map of normalized expression of Th1 signature genes in sorted WT or *Myd88−/−* CD4+ T cells: upregulated in red, downregulated in blue. (**C**) Gene Set Enrichment Analysis (GSEA) applied to genes highly expressed in WT or *Myd88−/−* CD4+ T cells using gene signatures of naïve and memory CD4+ T cells. Gene sets from naïve, central memory (CM), effector memory (EM) CD4+ T cells were generated using
*Figure 3 continued on next page*

*Figure 3 continued*

the top 200 differentially expressed genes in each subset (see Methods). Gene sets enriched in WT cells (red bars) or in *Myd88*−/− cells (blue bars) are displayed. (**D**) GSEA using Selected BioCarta (top) and Reactome (bottom) pathways (nominal p-value<0.05) enriched in WT cells (red bars) or in *Myd88*−/− cells (blue bars) are displayed. GSEA was performed using the pre-ranked genes and parameters as described in (**C**). Colors represent higher expression in WT (red) or *Myd88*−/− (blue). Gene networks with genes *in core* from GSEA analysis are shown in *Figure 3—figure supplement 1*.

DOI: https://doi.org/10.7554/eLife.30883.006

The following source data and figure supplement are available for figure 3:

**Source data 1.** Table 1. Differential expressed genes between WT and *Myd88*−/− CD4+ T cells sorted from infected BM-mixed chimeras.

DOI: https://doi.org/10.7554/eLife.30883.008

**Figure supplement 1.** Gene networks with the *genes in core* from GSEA analysis.

DOI: https://doi.org/10.7554/eLife.30883.007

18R1 is expressed mainly on CD44hiCD62L− (effector and EM) CD4+ T cells of infected mice, while IL-1R1 is mainly expressed on CD44hiCD62L+ (CM) CD4+ T cells. Note that no difference on the percentage of IL-1R1+ cells among CM CD4+ T cells was found between infected and non-infected control mice. These results indicated IL-18R, but not IL-1R, signaling as an important event in the generation of the Th1 cell response to infection.

To further test the hypothesis that the impaired expansion of IFN-γ-producing cells among *Myd88*−/−CD4+ T cells is due to the lack of IL-18R signaling, we generated mixed BM chimeras by the irradiation of WT B6 x B6.SJL F1 mice followed by reconstitution with a 1:1 mix of WT B6.SJL and *Il1r1*−/−, or *Il18r1*−/−, or *Myd88*−/− BM cells; control WT:WT→WT chimeras, which were reconstituted with a mix of B6.SJL and B6 BM were also generated. As shown in *Figure 5*, the percentage and the absolute numbers of IFN-γ+ cells among *Il18r1*−/−CD4+ T lymphocytes in mixed WT:*Il18r1*−/− chimeras and among *Myd88*−/−CD4+ T lymphocytes in mixed WT:*Myd88*−/− chimeric mice were significantly lower than the levels observed for WT CD4+ T cells in both sets of chimeras; the same was not true for *Il1r1*−/− IFN-γ+CD4+T cells, which were present at similar percentages and absolute numbers to WT IFN-γ+CD4+ T cells in WT:*Il1r1*−/− chimeric mice. We then analyzed BrdU incorporation *in vivo* by CD4+ T cells in the same groups of infected chimeras. As shown in *Figure 6A and B*, again, results obtained in WT:*Il18r1*−/−, but not in WT:*Il1r1*−/− chimeras, reflected the results obtained in WT:*Myd88*−/− chimeric mice (gate strategy on *Figure 6—figure supplement 1*). The same results were obtained when we analyzed the percentage and absolute numbers of cells expressing the Th1-associated chemokine receptor CCR5 among WT, *Myd88*−/−, *Il1r1*−/− or *Il18r1*−/− CD4+ T lymphocytes (*Figure 6C and D*, gate strategy on *Figure 6—figure supplement 2*), confirming the results obtained in microarray analyzes of WT and *Myd88*−/− CD4+ T cells (*Figure 3*).

On the other hand, the lack of MyD88 or IL-18R signaling has a lower impact in impairing the expansion of IFN-γ+CD8+ T cells (*Figure 1E and H* and *Figure 6E and F*, gate strategy on *Figure 6—figure supplement 3*), compared to its effect on the CD4+subset (*Figure 1D and G* and *Figure 5*). Moreover, when the levels of granzyme B+ (GzB+) CD8+ T cells were compared between the WT and KO compartments in the chimeric mice, no difference in the percentage nor in the absolute numbers of CD8+CTLs was found between WT and KO cells in any of the different mixed BM chimeras (*Figure 6G and H*, gate strategy on *Figure 6—figure supplement 4*), in accordance with our previous study showing that the cytotoxic response mediated by CD8+ T cells is not affected in *Myd88*−/− mice (*Oliveira et al., 2010*). We then analyzed the expression of the activation/memory marker CD44 on CD4+ T cells and found that the lack of expression of MyD88, or IL-1R, or IL-18R reduced the percentages and numbers of CD44hiCD4+ T cells (*Figure 6I and J*, gate strategy on *Figure 6—figure supplement 5*). The lower percentages of CD44high among KO-derived CD4+ T cell are in agreement with the our GSEA analysis of microarray data, which indicates the upregulation of genes related to central and effector memory phenotypes in WT but not in *Myd88*−/−CD4+ T cells (*Figure 3C*). Together, these results show that CD4+ T lymphocyte-intrinsic IL-18R/MyD88 signaling is required for the full differentiation and/or expansion of Th1 cells.

## *Il18r1*−/−mice are highly susceptible to infection with *T. cruzi*

To verify the impact of the absence of IL-18R- or MyD88-mediated signaling on resistance to infection, we compared parasitemia levels, as well as survival and parasite load in the myocardium, between *Il18r1*−/−, *Myd88*−/− and WT (B6) mice (*Figure 7A–C*). As shown in *Figure 7A*, WT mice

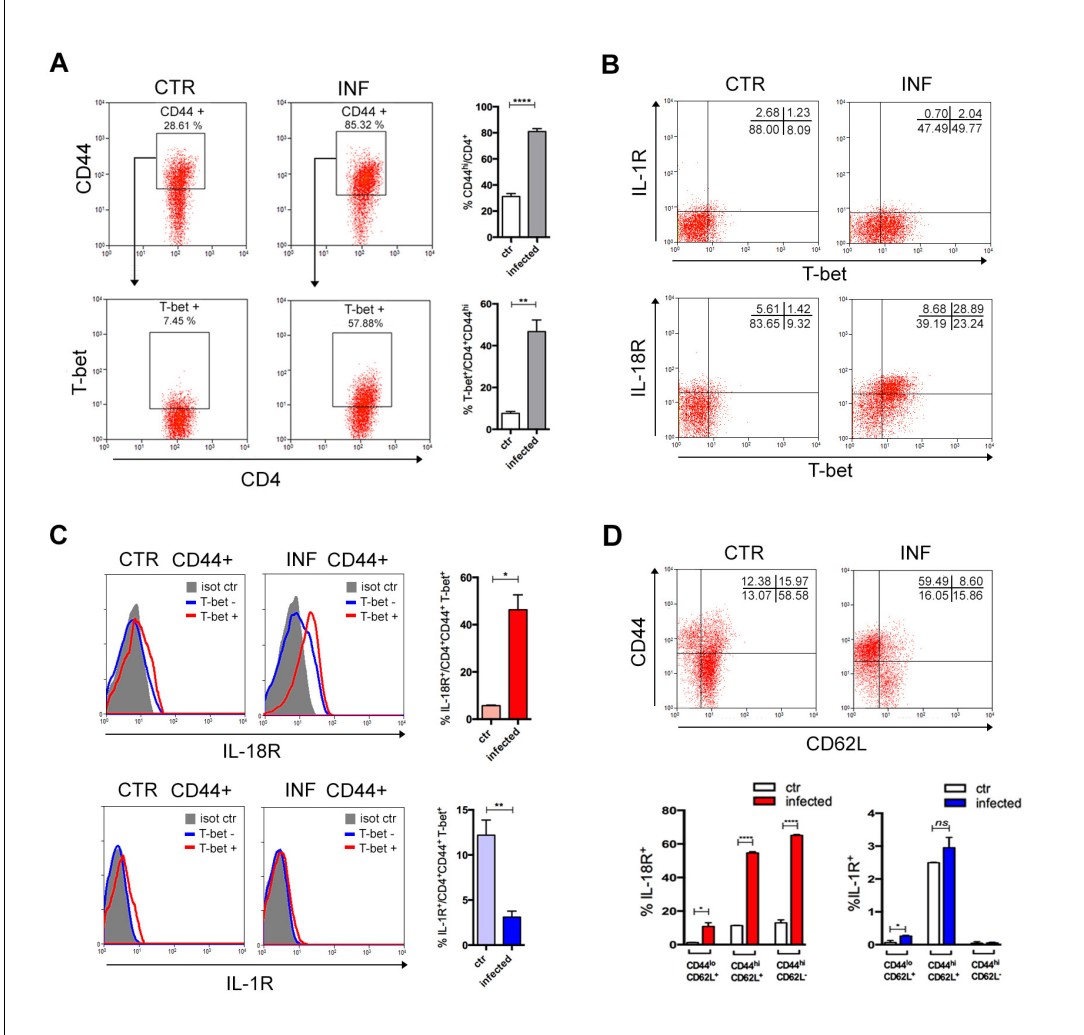

**Figure 4.** IL-18R, but not IL-1R, is upregulated in CD4+T cells upon activation following infection. (**A**) Gate strategy and mean percentage of CD44hiCD4+ and T-bet+CD44hiCD4+ cells gated on CD4+ T splenocytes from non-infected controls and infected B6 mice at day 14 pi. (**B**) Representative dot plot of IL-1R, IL-18R and T-bet expression, gated on CD44hiCD4+ T cells as in (**A**). (**C**) Representative histograms and mean percentages of IL-1R and IL-18R expression, gated on T-bet+CD44hiCD4+ T cells, as shown in (**A**) and (**B**). (**D**) Representative dot plot of CD44 and CD62L expression, gated on CD4+ T cells and mean percentages of IL-1R and IL-18R expression on CD44loCD62L+, CD44hiCD62L- and CD44hiCD62L+ gated on CD4+ T cells (n = 3 to 4). Error bars = SEM, *p≤0.05; **p≤0.01; ****p≤0.0001; ns = non significant (two-tailed Student *t*-test). Data are representative of 3 independent experiments. Kinetics of IL-1β and IL-18 levels in the serum, as well as of the appearance of IFN-γ+CD4+ T cells in the spleen are shown in *Figure 4—figure supplement 1*.

DOI: https://doi.org/10.7554/eLife.30883.009

The following figure supplement is available for figure 4:

**Figure supplement 1.** Kinetics of IL-1β and IL-18 levels in the serum, as well as of the appearance of IFN-γ+CD4+T cells in the spleen of infected mice.

DOI: https://doi.org/10.7554/eLife.30883.010

presented significantly lower parasitemia when compared to either *Il18r1*−/− or *Myd88*−/− mice at day 9 pi, when the peak of parasites in the blood is attained, and no statistical difference in parasitemia levels was found between *Il18r1*−/− and *Myd88*−/− mice. On the other hand, *Myd88*−/− mice are the most susceptible to infection, as all mice of this strain were dead by day 15 pi, while the totality of *Il18r1*−/− mice were dead only by day 23 pi, a time point at which only 32.5% of WT mice have succumbed (*Figure 7B*). Parasite loads were measured in the myocardium by qPCR at day 14 pi and a significantly higher load was found in *Myd88*−/− compared to *Il18r1*−/− mice, which in turn also display a higher parasite load in the heart when compared to WT mice (*Figure 7C*). This result is in accordance with the earlier mortality observed in *Myd88*−/− mice and might reflect the importance

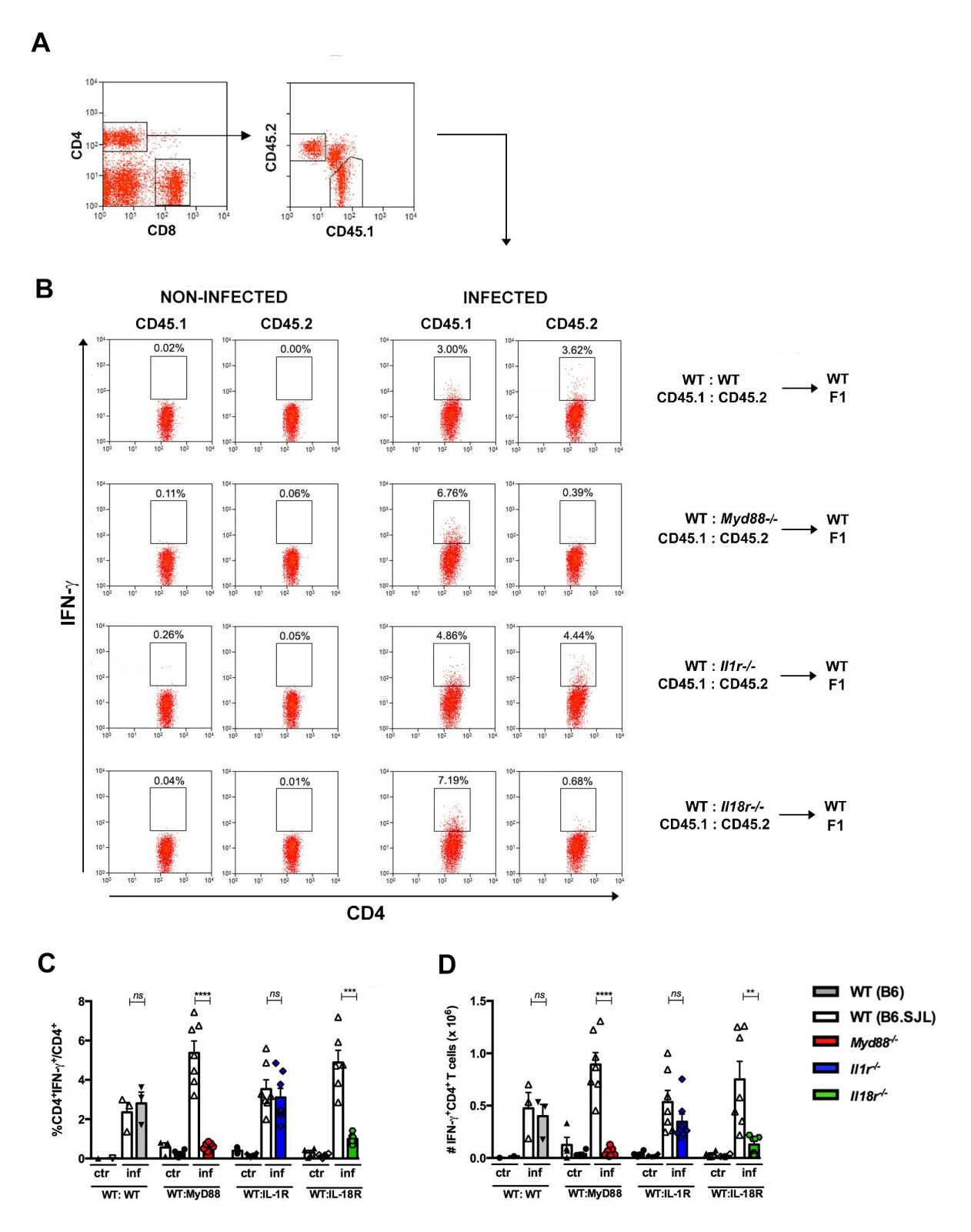

**Figure 5.** Effects of the absence of T cell-intrinsic MyD88, IL-1R or IL-18R signaling on IFN-γ production by spleen CD4[+] T cells of infected mixed BM chimeras. (**A**) Representative dot plots showing gating strategy for CD4[+] T cells and for CD45.1[+] (B6.SJL WT) or CD45.2[+] (B6 WT or *Myd88*[−/−] or *Il1r1*[−/−] or *Il18r1*[−/−]) splenocytes from mixed BM chimeras. (**B**) Representative dot plots of IFN-γ expression gated on CD4[+]CD45.1[+] (B6.SJL WT) or CD4[+]CD45.2[+] (B6 WT or *Myd88*[−/−] or *Il1r1*[−/−] or *Il18r1*[−/−]) splenocytes from non-infected control or infected (2 × 10[3] blood trypomastigotes, ip) mixed
*Figure 5 continued on next page*

Figure 5 continued

BM chimeras, on day 14 pi. (C) Percentages and (D) absolute cell numbers of IFN-γ⁺CD4⁺ T cells gated as in (A) and (B). Bars are the mean of combined data from 2 independent experiments (n = 3 to 7 individually analyzed chimeric mice); error bars = SEM; ns = non significant; **p≤0.01; ***p≤0.001, ****p≤0.0001 (two-tailed Student *t*-test). Data are representative of 3 to 4 independent experiments.

DOI: https://doi.org/10.7554/eLife.30883.011

of innate signaling through TLRs in the tissue for parasite control (*Rodrigues et al., 2012*). We then analyzed parameters of the acquired response in the three strains of mice. Notably, both *Il18r1*⁻/⁻ and *Myd88*⁻/⁻ mice presented lower percentages and absolute numbers of IFN-γ⁺CD4⁺ T cells in the spleen, when compared to WT mice (*Figure 7D and G*), but the percentages of IFN-γ⁺CD8⁺ and GzB⁺CD8⁺ T cells were found to be equivalent in the three mouse strains (*Figure 7E,F,H and I*). In summary, *Il18r1*⁻/⁻ mice presented an intermediary level of susceptibility to infection with *T. cruzi*, in terms of survival and parasite load in the myocardium, when compared to *Myd88*⁻/⁻ and WT (B6) mice.

## Adoptive transfer of WT CD4⁺cells to *Il18r1*⁻/⁻ and to *Myd88*⁻/⁻ mice increases resistance to infection

Given that both *Il18r1*⁻/⁻ and *Myd88*⁻/⁻ infected mice displayed lower levels of Th1 cells in the spleen, when compared to WT animals, we asked whether the adoptive transfer of WT CD4⁺ T lymphocytes to *Il18r1*⁻/⁻ and *Myd88*⁻/⁻ strains would be enough for lowering parasitemia levels and improve survival to infection, in accord with the crucial role of Th1 responses for protection against *T. cruzi* infection (*Michailowsky et al., 2001*). For this, CD4⁺ T cells were sorted from infected WT mice to high purity levels (*Figure 8—figure supplement 1A*) and were then transferred to both *Il18r1*⁻/⁻ and *Myd88*⁻/⁻ mice, 20 hr before infection. As shown in *Figure 8*, this was in fact the case: WT CD4⁺ T cells significantly improved control of parasite levels (*Figure 8B and D*) and survival (*Figure 8A and C*) in both strains. Moreover, the ability to lower parasitemia required the capacity of secreting IFN-γ by transferred CD4⁺ T cells, since the adoptive transfer of *Ifng*⁻/⁻ CD4⁺ T cells to *Myd88*⁻/⁻ mice had no effect (*Figure 8—figure supplement 1B*). However, while the mortality rate in *Il18r1*⁻/⁻ mice that received WT CD4⁺ T cells attained the rate observed in WT B6 mice (*Figure 8A*), the adoptive transfer of WT CD4⁺ T cells into *Myd88*⁻/⁻ mice ameliorated survival only partially (*Figure 8C*). The same result was obtained when transferring 4 × 10⁶ purified WT CD4⁺ T cells, that is, 3 times more cells than in the experiment shown in *Figure 8* (data not shown). Note, however, that the transfer of WT CD4⁺ T cells reduced parasitemia to numbers observed in WT mice, both in *Il18r1*⁻/⁻ and in *Myd88*⁻/⁻ mice, as shown in *Figure 8B and D*. Together, these results show that while increasing the numbers of IFN-γ⁺CD4⁺ T cells in *Il18r1*⁻/⁻ mice is enough for restoring their resistance to infection to the WT level, both in terms of parasitemia and survival, this is not the case for *Myd88*⁻/⁻ mice. Although in transferred *Myd88*⁻/⁻ mice parasitemia was decreased to the WT level (as in transferred *Il18r1*⁻/⁻mice) and survival was also significantly delayed, still, the totality of *Myd88*⁻/⁻ mice succumbed to infection.

## Discussion

Here, we have shown that T cell-intrinsic IL-18R/MyD88 signaling is crucial for the development of a robust Th1 response induced by infection with the intracellular pathogen, *T. cruzi*. Although previous studies, in different models of infection, have shown the importance of MyD88 signaling intrinsic to T cells, none of those works has identified the receptor upstream the MyD88 adaptor molecule playing a determinant role for the Th1 response *in vivo* (*Frazer et al., 2013*; *LaRosa et al., 2008*; *Raetz et al., 2013*; *Zhou et al., 2009*). IL-18 has been reported as a relevant factor in protective immunity against several intracellular pathogens and its functions *in vivo* are multiple, being produced by and acting on different cell types (*Nakanishi et al., 2001*; *Garlanda et al., 2013*). The role of IL-18 in Th1 differentiation has been however neglected, maybe because it has been shown dispensable for Th1 development under certain circumstances, as for example when IL-12 is present in high levels, (*Haring and Harty, 2009*; *Monteforte et al., 2000*). IL-18 synergizes with IL-12 in promoting IFN-γ secretion and Th1 differentiation, but it is not capable of promoting Th1 polarization *per se* and it is thus considered a reinforcing or stabilizing factor for Th1 commitment

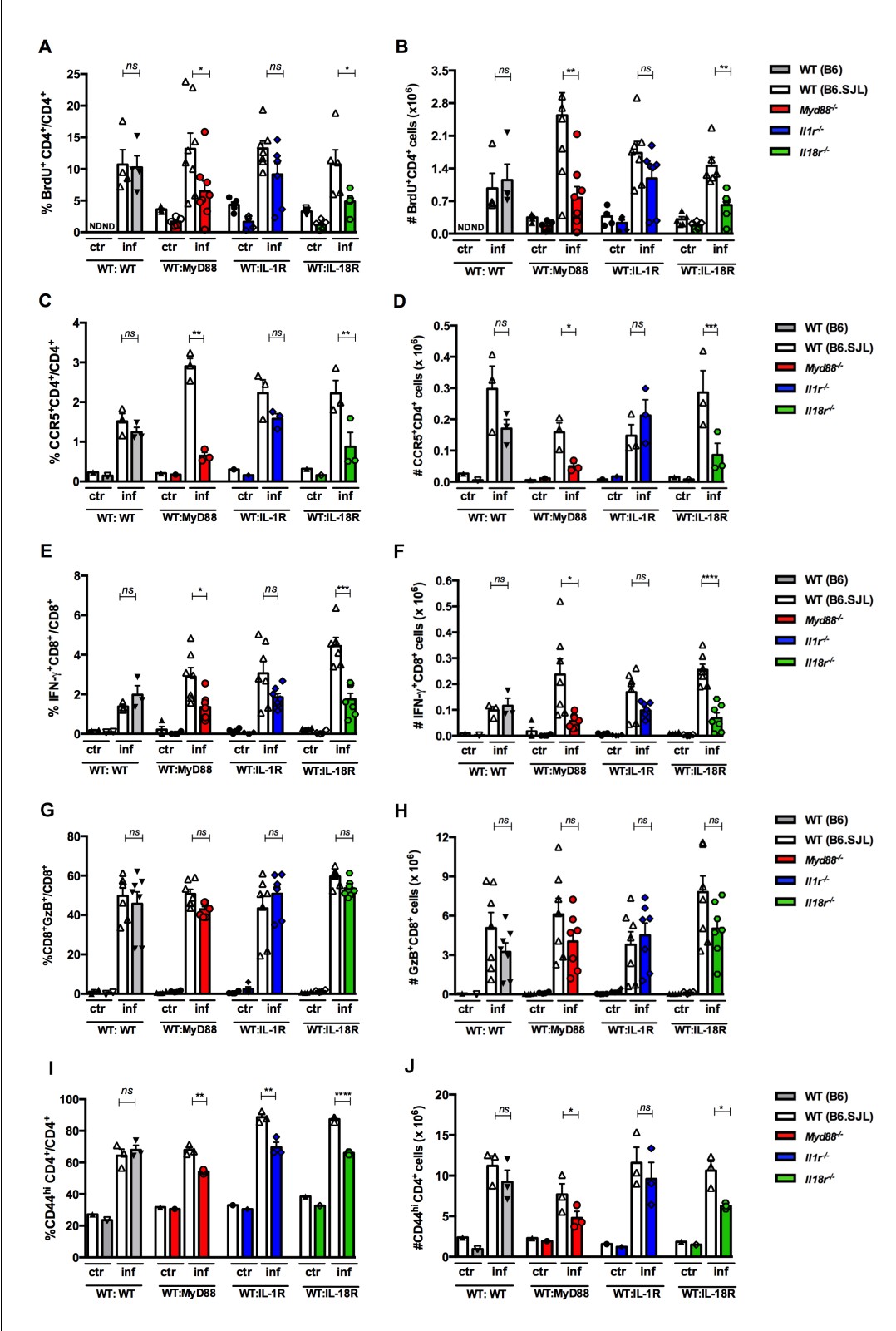

**Figure 6.** Effects of the absence of T cell-intrinsic MyD88, IL-1R or IL-18R signaling on the proliferation and phenotype of CD4$^+$ and CD8$^+$spleen T cells from infected mixed BM chimeras. (A, C, G, E and I) Mean frequencies and (B, D, F, H and J) absolute cell numbers of: (A and B) BrdU$^+$CD4$^+$ and (C and D) CCR5$^+$CD4$^+$ cells gated on WT CD45.1$^+$ or KO CD45.2$^+$ CD4$^+$ T cells, as in *Figure 5A*. (E and F) IFN-γ$^+$CD8$^+$ and (G and H) GzB$^+$CD8$^+$ T cells gated on WT CD45.1$^+$ or KO CD45.2$^+$ CD8$^+$ T cells, as shown in *Figure 6—figure supplement 3*. (I and J) CD44$^{hi}$CD4$^+$ T cells, gated on WT or KO

*Figure 6 continued on next page*

*Figure 6 continued*

CD4$^+$ T cells as in *Figure 5A*. Bars are the mean of individually analyzed mice from one (C, D, I and J) or two combined independent experiments (A, B, E, F, G and H). (n = 3 to 8); error bars = SEM; ND = not done; ns = non significant; *p≤0.05; **p≤0.01; ***p≤0.001, ****p≤0.0001 (two-tailed Student *t*-test). Gate strategies and representative dot plots are shown in *Figure 6—figure supplements 1–5*.

DOI: https://doi.org/10.7554/eLife.30883.012

The following figure supplements are available for figure 6:

**Figure supplement 1.** Representative dot plots and gate strategy of BrdU$^+$CD4$^+$T cell analysis from chimeric mice.

DOI: https://doi.org/10.7554/eLife.30883.013

**Figure supplement 2.** Representative dot plots and gate strategy of CCR5$^+$CD4$^+$ T cell analysis from chimeric mice.

DOI: https://doi.org/10.7554/eLife.30883.014

**Figure supplement 3.** Representative dot plots and gate strategy of IFN-γ$^+$CD8$^+$ T cell analysis from chimeric mice.

DOI: https://doi.org/10.7554/eLife.30883.015

**Figure supplement 4.** Representative dot plots and gate strategy of GzB$^+$CD8$^+$ T cell analysis from chimeric mice.

DOI: https://doi.org/10.7554/eLife.30883.016

**Figure supplement 5.** Representative dot plots and gate strategy of CD44$^{hi}$CD4$^+$ T cell analysis from chimeric mice.

DOI: https://doi.org/10.7554/eLife.30883.017

(*Robinson et al., 1997*) and reviewed in *Berenson et al., 2004*). Early in the process of Th1 differentiation, IFN-γ is required to induce nuclear factors as T-bet and IRF1 and it is necessary for the expression of the IL-12 receptor β1 subunit (IL-12Rβ1) (*Kano et al., 2008*). Then, IL-12 signaling drives the up-regulation of IL-18Rα expression, which can next be maximized by IL-18 itself (*Smeltz et al., 2002*; *Berenson et al., 2004*).

GSEA analysis of our microarray data revealed that sets of genes involved in proliferation and protection from apoptosis were upregulated in WT in comparison to their expression in *Myd88*$^{-/-}$CD4$^+$ T cells. These results are in accordance with the preferential expansion of WT Th1 cells in vivo, measured by flow cytometry and *in vivo* BrdU incorporation. Also, flow cytometry analyses of characteristic Th1 genes showed that percentages and absolute numbers of cells expressing IFN-γ and CCR5 among *Myd88*$^{-/-}$ and *Il18r1*$^{-/-}$CD4$^+$ T lymphocytes were found similarly lower when compared to WT cells in BM mixed chimeric mice. Therefore, together, these results indicate that the lack of intrinsic IL-18R signaling in CD4$^+$ T cells phenocopied the absence of MyD88 signaling, resulting in lower Th1 cell numbers *in vivo*.

It is worth noting that the majority of our analyses were performed at the peak of detection of the Th1 response in our system, day 14 pi, when the parasite is still present in the host (*Oliveira et al., 2010*). In this aspect, our present study differs from a recent report where the importance of IL-18R/MyD88 signaling for a noncognate Th1 response was put in evidence at later time points of infection with *Salmonella*, by the injection of LPS, (*O'Donnell et al., 2014*), apparently in a TCR-independent way, as previously proposed (*Robinson et al., 1997*; *Berenson et al., 2004*). The hypothesis that other MyD88-dependent T-cell intrinsic signaling, as TLR-pathways, might also be relevant for the Th1 response in the present system is, in our opinion, unlikely. We and others have previously shown that *Tlr4*$^{-/-}$ as well as *Tlr2*$^{-/-}$ mice infected with *T. cruzi* display normal levels of CD4$^+$IFNγ$^+$ T cells (*Bafica et al., 2006*; *Oliveira et al., 2010*). The possibility that TLR7 or TLR9, would play an intrinsic role in T cells would most likely require the phagocytosis of the parasite by T cells, in order to allow parasite-derived RNA and DNA to reach endosomal TLRs, a non-reported fact in T cells. The hypothesis that other MyD88-mediated signaling pathways, as IFN-γ-induced IRF1 (*Negishi et al., 2006*), might be also contributing for Th1 response was not formally disproven here. However, we believe that our data, showing an insufficient expansion of IL-18R-defective Th1 cells, of the same magnitude as observed for MyD88-defective Th1 lymphocytes, even in an environment where WT APCs are abundant, argue in favor of a model where the lack of a strong Th1 response observed in *Myd88*$^{-/-}$ mice is mostly due to the absence of IL-18R signaling in CD4$^+$ T cells.

A recent report also detected the importance of T-cell intrinsic MyD88 signaling for the induction of functionally competent Th1 and memory CD4$^+$ T cells, using *Myd88*$^{FL/FL}$*Cd4-cre* mice immunized with antigen plus adjuvant (*Schenten et al., 2014*). This is in agreement with our GSEA analysis, showing that sets of genes related to central and effector memory were up-regulated in WT, but not in *Myd88*$^{-/-}$CD4$^+$ T cells. In the cited study, however, the responsible pathway upstream MyD88

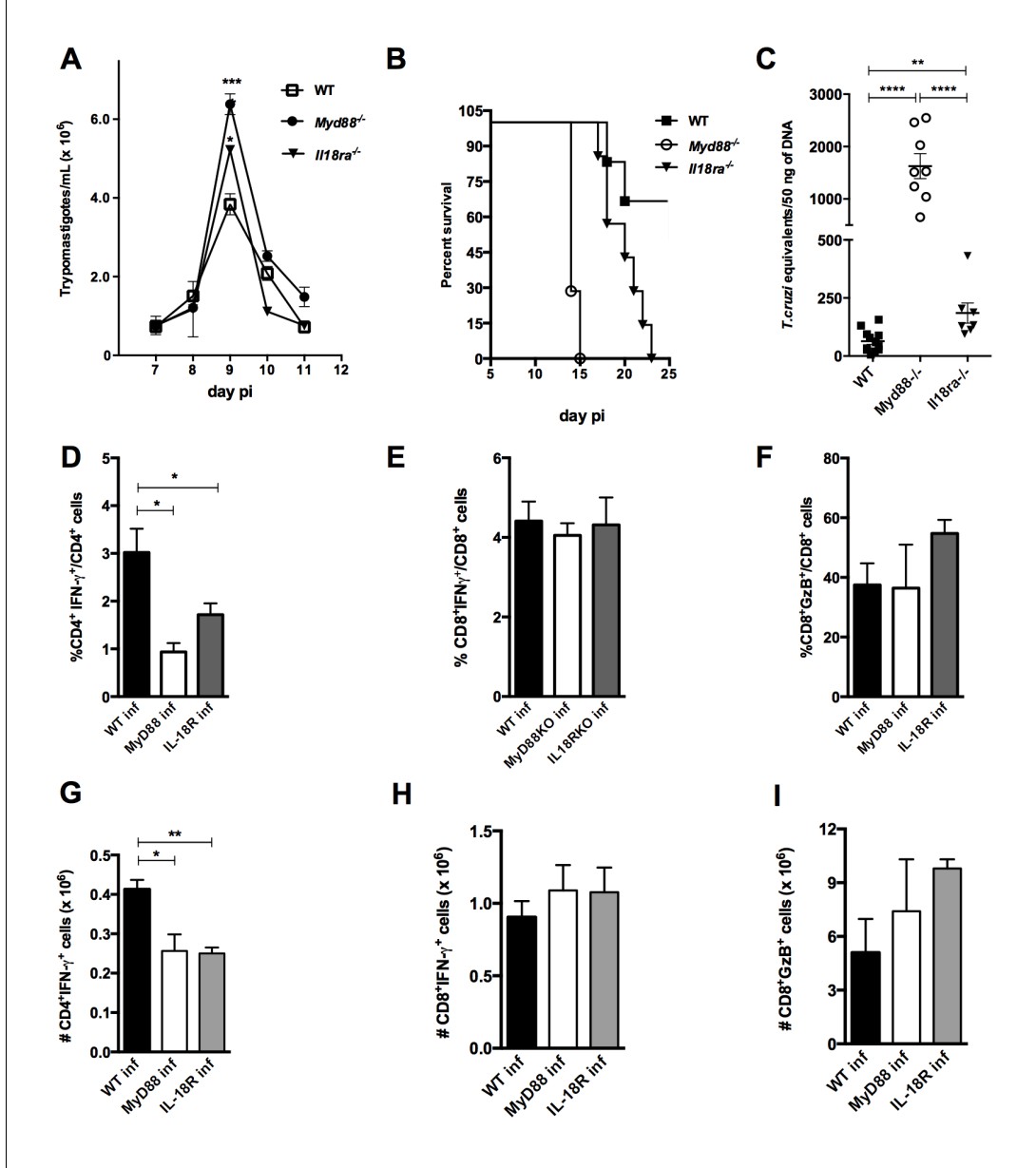

**Figure 7.** $Il18r1^{-/-}$ mice are highly susceptible to infection with *T. cruzi* and display lower frequency and absolute numbers of Th1 cells. (**A**) Parasitemia curve, (**B**) survival and (**C**) parasite load in the myocardium at day 14 pi of $Il18r1^{-/-}$, $Myd88^{-/-}$ and WT B6 mice infected with $2 \times 10^3$ blood trypomastigotes of the Y strain. Survival curves are statistically different (p<0.05). (**D–I**) Mean frequencies and absolute numbers of (**D, G**) IFN-$\gamma^+$CD4$^+$(**E, H**) IFN-$\gamma^+$CD8$^+$ and (**F, I**) GzB$^+$CD8$^+$ cells, gated on CD4$^+$ or CD8$^+$ T splenocytes from 4 to 8 individually analyzed mice at day 14 pi. Data are representative of 3 independent experiments. Error bars = SEM; ns = non significant; *p$\leq$0.05; **p$\leq$0.01; ****p$\leq$0.0001 (two-tailed Student *t*-test).

DOI: https://doi.org/10.7554/eLife.30883.018

was attributed to IL-1R, which was shown to render naive CD4$^+$ T cells refractory to Treg cell-mediated suppression. Although we have not studied the memory response in the infection model here, we also found a significant lower percentage of cells expressing high levels of the CD44 activation/memory marker among CD4$^+$ T cells lacking IL-1R expression, as well as among $Myd88^{-/-}$ and $Il18r1^{-/-}$CD4$^+$ T cells. Concerning the Th1 response, however, we could not detect any defect in IL-1R-deficient cells. Moreover, we found that contrary to IL-18R, IL-1R is not expressed on CD4$^+$ T cells in response to infection with *T. cruzi*. This apparent discrepancy between the cited study and

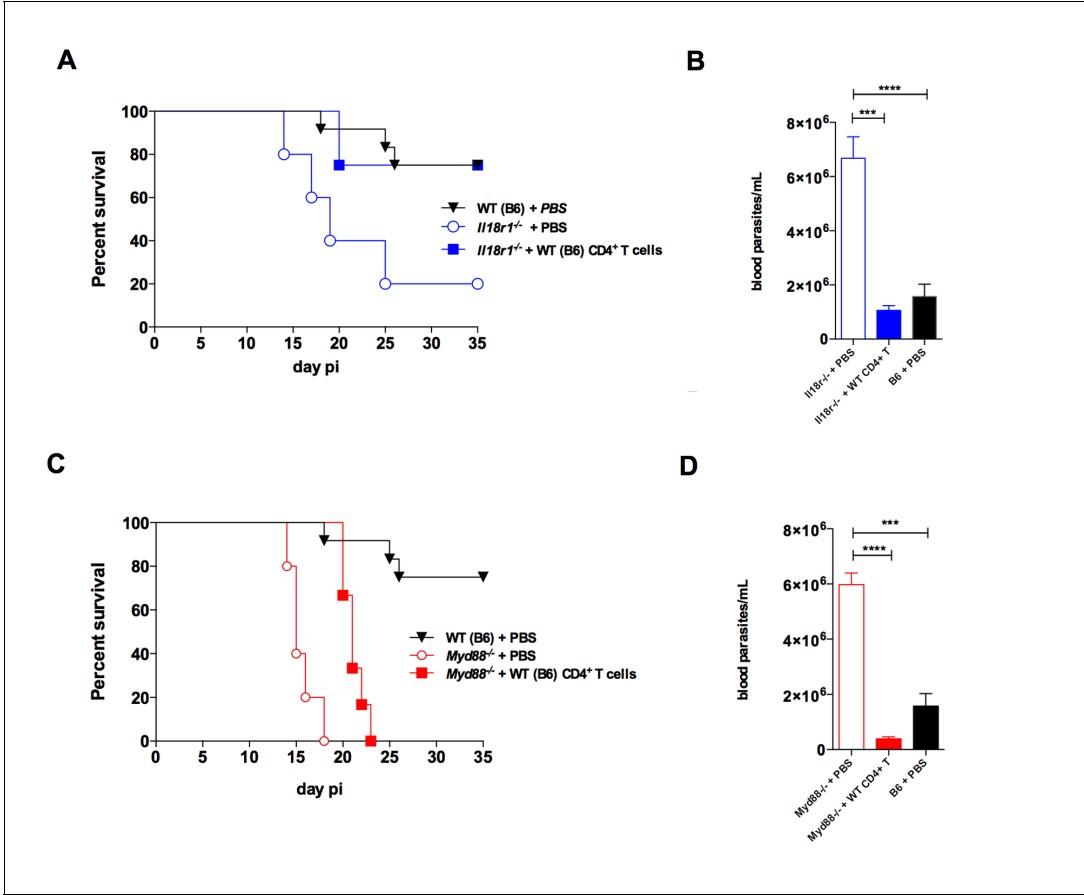

**Figure 8.** The adoptive transfer of purified WT CD4+ T cells protects *Il18r1*−/− and *Myd88*−/− mice against infection with *T. cruzi*. (**A and C**) Survival; (**B and D**) mean parasitemia levels at day 9 pi of *Il18r1*−/− (**A and B**) or *Myd88*−/− mice (**C and D**), which received or not 1.3 × 10^6 CD4+ T cells purified from infected WT B6 mice. Results obtained in infected but non-transferred WT B6 mice are shown in black lines and black bars. Data are representative of 2 independent experiments, using 8 to 10 male mice in each group; error bars = SEM. Survival curves of transferred and non-transferred mice are statistically different (p<0.05). Purity of adoptive transferred CD4+ T cells and parasitemia in *Myd88*−/− mice adoptively transferred with WT or *Ifng*−/− CD4+ T cells are shown in *Figure 8—figure supplement 1*.
DOI: https://doi.org/10.7554/eLife.30883.019

The following figure supplement is available for figure 8:

**Figure supplement 1.** Adoptive transfer of WT, but not of *Ifng*−/−CD4+ T cells diminished parasitemia in infected *Myd88*−/− mice.
DOI: https://doi.org/10.7554/eLife.30883.020

ours may be due to differences in the inflammatory response and cytokine milieu between an infection model and the Ag plus adjuvant system employed by Schenten and collaborators (*Schenten et al., 2014*). It is possible that diverse, higher and/or more prolonged cytokine levels are attained in the infection with *T. cruzi* compared to Ag plus adjuvant immunization. Possibly, during infection, another cytokine, such as IL-6, might be replacing the missing IL-1R signaling in T conventional cells to overcome Treg-mediated suppression, as previously described (*Nish et al., 2014*). It is also worthwhile to note that employing the system of mixed BM chimeras allowed us to compare WT and gene-deficient T cells competing with each other and being submitted to the same inflammatory milieu. Mixed BM chimera strategy also circumvents compensatory mechanisms observed in some knockout mouse strains.

We have also studied here the susceptibility of *Il18r1*−/− mice to infection with *T. cruzi*. This strain exhibited parasitemia as high as *Myd88*−/− mice and survival and parasite load in the myocardium at intermediary levels, between *Myd88*−/− and WT mice. Both *Il18r1*−/− and *Myd88*−/− mice have diminished levels of Th1 cells, but CD8+ T responses were not affected, as predicted by our previous work showing intact in vivo Ag-specific cytotoxicity in *Myd88*−/− mice (*Oliveira et al., 2010*).

Although $Il18^{-/-}$ mice were shown to be resistant to *T. cruzi* (*Graefe et al., 2003*), discrepancy between the outcomes in $Il18r1^{-/-}$ versus $Il18^{-/-}$ mice was also previously documented for infection with *M. tuberculosis* and in autoimmunity (*Gutcher et al., 2006*; *Schneider et al., 2010*). The reason for this fact is not clear so far, but a reported compensatory mechanism increasing IL-12 production in $Il18^{-/-}$ mice may explain the observed higher resistance of this strain (*Monteforte et al., 2000*). Nevertheless, we clearly demonstrated here that $Il18r1^{-/-}$ mice display low levels of IFN-$\gamma^+$CD4$^+$ T cells in response to infection with *T. cruzi* and are highly susceptible to infection. These results are consistent with our findings demonstrating the essential role of T cell-intrinsic IL-18R/MyD88 signaling for cognate Th1 response to this parasite. Furthermore, the adoptive transfer of WT CD4$^+$ T cells was able to reduce parasitemia to WT levels in both $Il18r1^{-/-}$ and $Myd88^{-/-}$ infected mice, in an IFN-$\gamma$-dependent way, and to improve survival in both strain. However, mortality was brought to WT rate only in the adoptively transferred $Il18r1^{-/-}$ strain. These results are in accordance with the fact that mortality due to infection with *T. cruzi* is not always a simple and direct correlation to parasitemia. Instead, in some cases, mortality is due to increased pro-inflammatory cytokine release (*Hölscher et al., 2000*). It is tempting to speculate that other MyD88-independent innate mechanisms, as the TLR4/TRIF pathway, might be responsible for this putative pro-inflammatory effect in $Myd88^{-/-}$ mice. Alternatively, but not mutually exclusive, tissue-tolerance mechanisms, possibly involving MyD88-dependent feedback control responses, might protect the host from immune- or pathogen-inflicted damage and would be impaired in $Myd88^{-/-}$ mice. This hypothesis is presently being investigated in our laboratory.

In conclusion, our work unequivocally demonstrates that the absence of IL-18R exclusively in T cells during infection leads to a deficient cognate Th1 response. By excluding other cell types, as NK cells and the IL-1R receptor-signaling pathway in this process, our work brings a conceptual advance to the field. Adding to the established paradigm of TLR/MyD88-dependent activation of APCs for IL-12 production and the resulting initiation of Th1 response, our data demonstrated the critical role of T cell-intrinsic IL-18R/MyD88 signaling for the reinforcement of Th1 response in vivo, placing MyD88-dependent signaling pathways on the control of multiple phases of Th1 response: co-stimulation, differentiation, expansion and memory establishment. Altogether, the results presented here disclose and extend a mechanistic framework for understanding CD4$^+$ T cell responses, whose optimization may provide new therapeutic and vaccination strategies to combat intracellular pathogens.

## Materials and methods

### Mice and ethics statement

C57BL/6, B6.SJL and (B6 x B6.SJL) F1 mice were obtained from the Universidade Federal Fluminense (UFF, Niterói, Brazil). $Myd88^{-/-}$ mice were generated by Dr. S. Akira (Osaka University, Japan) and backcrossed onto C57BL/6 genetic background for 9 generations; $Rag2^{-/-}$ and $Il18r1^{-/-}$ mice were donated by Dr. R. Gazzinelli (UFMG, MG, Brazil) and Dr. B. Ryffel (CNRS, Orleans, France), respectively. $Il1r1^{-/-}$ mice were purchased from JAX Mice, USA. Experiments were conducted in accordance with guidelines of the Animal Care and Use Committee of the Federal University of Rio de Janeiro (Comitê de Ética do Centro de Ciências da Saúde CEUA-CCS/UFRJ). Procedures and animal protocols were approved by CEUA-CCS/UFRJ license n.: IMPPG022.

### Experimental infection and parasite load

Five to six week-old C57BL/6, $Myd88^{-/-}$ or $Il18r1^{-/-}$ male mice, or BM mixed chimeric mice after 6–8 week of reconstitution were inoculated intraperitoneally (ip) with $2 \times 10^3$ trypomastigotes/0.2 ml bloodstream trypomastigotes of the Y strain of *T. cruzi*. Parasitemia was monitored by counting the number of bloodstream trypomastigotes in 5 µl of fresh blood collected from the tail vein and mouse survival was followed daily. Hearts of infected mice were excised after perfusion, minced and the cardiac tissue immediately homogenized in 1.0 ml of 4.0 M Guanidine thiocyanate (SIGMA-Aldrich) containing 8.0 µl/ml of β-mercaptoethanol (SIGMA-Aldrich) and processed for DNA extraction. Generation of PCR standards and detection of parasite tissue load by real-time PCR was carried out as described (*Cummings and Tarleton, 2003*); briefly, primers amplify a repeated satellite sequence of *T. cruzi* DNA of 195 base-pairs: TCZ-Fwd.: (5'-GCTCTTGCCCACAAGGGTGC-3') and TCZ-Rev.: (5'-CCAAGCAGCGGATAGTTCAGG-3'). Reactions with TNF-α-Fwd: (5'- CC

TGGAGGAGAAGAGGAAAGAGA-3') and TNF-α-Rev.: (5'- TTGAGGACCTCTGTGTATTTGTCAA-3') primers for *Mus musculus* TNF-α gene were used as loading controls. PCR amplifications were analyzed using primers in combination with SYBR Green on a StepOne Real Time PCR System (Applied Biosystems, Life Technologies).

## Reagents and antibodies
See table below.

## Generation of BM chimeras and *Rag2*$^{-/-}$ reconstituted mice
For BM chimeras, acidic drinking water (pH 2,5–3,0) was given for 8 days to B6 x B6.SJL F1 (CD45.2$^+$ CD45.1$^+$) recipient mice prior to irradiation with a lethal single dose of 800 rad (TH780C - Theratronics, Canada).The day after irradiation, BM cells were transferred and mice were treated with neomycin sulfate (2 mg/mL) in the drinking water for 15 days. Marrow was harvested from the femur and tibia of B6.SJL (CD45.1$^+$) and B6, *Myd88*$^{-/-}$, *Il1r1*$^{-/-}$ or *Il18r1*$^{-/-}$ (CD45.2$^+$) mice by flushing with cold RPMI media (GIBCO). BM cells were washed thoroughly with PBS and $4 \times 10^6$ total cells were injected into the irradiated recipient mice. For mixed chimeras, CD45.1$^+$ and CD45.2$^+$ BM cells were transferred in a 1:1 ratio ($2 \times 10^6$ cells each). Alternatively, non-irradiated mixed chimeric *Rag2*$^{-/-}$ mice were generated by the *iv* injection of $20 \times 10^6$ spleen cells from WT B6.SJL (CD45.1$^+$) and *Myd88*$^{-/-}$ (CD45.2$^+$) mice in a 1:1 ratio. Six to eight weeks after the first reconstitution, *Rag2*$^{-/-}$ mice received a second injection of WT B6 x B6.SJL F1 (CD45.1$^+$CD45.2$^+$) and *Myd88*$^{-/-}$ (CD45.2$^+$) splenocytes in a 1:1 ratio, 6 hr before being infected. In all experiments, mice were infected 6 to 8 weeks after cell transfer, when reconstitution was achieved.

## Flow cytometry
On different days post infection, mice were euthanized and spleens were harvested. Spleen cells were treated with ACK buffer for red blood cell lysis, washed and then stained. Single cell suspensions from spleen were incubated with anti-CD16/CD32 (FcR block) for 5 min and then stained with anti-CD45.1, anti-CD45.2, anti-CD11c, anti-CD3, anti-CD4, anti-CD8, anti-CCR5, anti-TCRβ, anti-T-bet, anti-IL-1R1, anti-IL-18R1, anti-CD62L or anti-CD44 antibodies (see table below) for surface staining for 30 min on ice. Alternatively, $2 \times 10^6$ spleen cells were cultured in the presence of monensin (5 μM) and 2 μM of K$^b$-restricted Tskb20 peptide (Genscript) for 10 hr. After staining of surface markers, cells were fixed with paraformaldehyde 1% for 1 hr and permeabilized with saponin 0.2% for 20 min. At least 20,000 events gated on CD4$^+$ T lymphocytes were acquired. Analytical flow cytometry was conducted with a MoFlo (Beckman Coulter/Dako-Cytomation) or FACSCalibur (BD Bioscience) and the data were analyzed with Summit V4.3 software (Dako Colorado, Inc).

## *In vivo* BrdU incorporation assay
Mixed chimeric mice were infected with *T. cruzi* and BrdU (1.0 mg/animal) was administered ip at days 12 and 13pi every 12 hr. At day 14 pi, mice were sacrificed and $2 \times 10^6$ spleen cells were surface stained with anti-CD45.1, anti-CD45.2, anti-CD3, anti-CD4 and anti-CD8 (see table below). For BrdU analysis, after surface staining, cells were fixed and permeabilized with 2% paraformaldehyde, 0.02% Tween 20 in PBS overnight at 4°C. Then, cells were washed with PBS and incubated in DNase buffer (150 mM NaCl, 5 mM MgCl$_2$, 10 μM HCl, 100KU/mL DNase I) for 1 hr at RT. Subsequently, cells were washed with PBS, 2% FCS, 0.05% sodium azide and stained with FITC-conjugated anti-BrdU for 30 min at RT.

## ELISA
Blood samples were collected at different days of infection and the sera were stored at - 80°C. Samples from individual mice were diluted 10x for IL-1β quantification or non-diluted, for IL-18 detection. Cytokine levels were determined using mouse IL-18/IL-1F4 ELISA kit and mouse IL-1beta/IL-1F2 DuoSet ELISA kit (R and D Systems) according to the manufacturer's instructions.

## Adoptive CD4$^+$ T cell transfer assay
Eight week-old male C57BL/6 mice were infected with 10$^4$ trypomastigotes and at day 9 pi spleen cells were harvested and treated with ACK for red blood cells lysis. For CD4$^+$ T cell enrichment by

negative selection, $3 \times 10^8$ spleen cells were labeled with anti-B220, anti-CD11b, anti-NK1.1, anti-$\gamma\delta$ and anti-CD8 biotinylated antibodies (see table below) for 30 min on ice and incubated with Dynabeads® Biotin Binder (Invitrogen, Life Technology) for 30 min on ice under agitation and depleted, according to manufacturer's instructions. The CD4$^+$-enriched cells were stained with PE-conjugated anti-CD4, and PE-Cy7-conjugated Streptavidin. For sorting, PE-Cy7$^+$ cells were excluded and CD4$^+$ T lymphocytes were sorted using MoFlo (Beckman Coulter/Dako-Cytomation) flow cytometer. CD4$^+$ T cells presenting purity higher than 98% were adoptively transferred by *iv* injection into *Myd88*$^{-/-}$ or *Il18r1*$^{-/-}$ recipient mice ($1.3 \times 10^6$ cells/animal) followed by infection with $2 \times 10^3$ trypomastigotes, 20 hr later. Parasitemia and survival were accompanied daily.

## Gene-expression profiling

Mixed BM chimeric WT(B6.SJL):*Myd88*$^{-/-}$→ WT (B6 x B6.SJL F1) mice were euthanized at day 14 pi, spleen cells were collected and the enrichment of CD4$^+$ cells was done as described above. Cells were stained with PB-anti-CD45.1, APC-Cy7-anti-CD45.2 and PE-anti-CD4 antibodies (see table below) and the sorting was conducted with a MoFlo (Beckman Coulter/Dako-Cytomation) flow cytometer. CD4$^+$CD45.1$^+$ cells (WT) and CD4$^+$CD45.2$^+$ cells (*Myd88*$^{-/-}$) were sorted (99% of purity) and RNA was prepared from sorted cell populations by Trizol (Invitrogen) extraction followed by RNeasy extraction (Qiagen), according to manufacturer's protocols. For microarray analysis, RNA was labeled and hybridized to Illumina MouseRef-8 v2.0 Expression BeadChips array according to the Illumina protocols by the Rockefeller University Genomic facility. Gene expression data were extracted with GeneSpring software GX 11.0 software using default parameters. Data was then normalized by subtracting the average from all genes from each gene intensity and dividing the result by the standard deviation of all gene intensities (i.e. z-score normalization) Z ratios were calculated as previously described (*Cheadle et al., 2003*) and used to compare the difference in expression between WT samples and *Myd88*$^{-/-}$ samples. Gene Set Enrichment Analysis (GSEA) (http://www.broadinstitute.org/gsea/index.jsp) was performed on the genes pre-ranked by Z ratios (1000 permutations and Weighted Enrichment Statistic) and using 3 different gene sets: BioCarta pathways (c2), Reactome pathways (c2) and GSE11057. GSE11057 gene set is comprised of genes highly expressed in 3 subsets of CD4$^+$ T cells: Central memory (CM), Effector memory (EM) and Naïve. To identify these genes, we re-analyzed the microarray data set from reference (*Abbas et al., 2009*), using the GEO2R tool (http://www.ncbi.nlm.nih.gov/geo/geo2r/) (Adj p-value<0.01; top 200 genes based on log2 fold-change on CM vs naïve and EM vs naïve comparisons). Interactions among the 'genes in core' from the GSEA analyses were assessed and visualized by Ingenuity Pathway Analysis (IPA) database.

## Statistical analysis

Statistical analyses were performed using GraphPad Prism version 5 for Windows (GraphPad Software, San Diego California USA, www.graphpad.com). Sample sizes were chosen to allow the detection of statistical differences with p<0.5 using a two-tailed Student's *t* test or the Log-rank (Mantel-Cox) test, for mouse survival curves after challenge with *T. cruzi*. When comparing two groups, the minimum number was n = 3 per group. In some experiments, the number was increased to n = 9 to permit p<0.5 even considering the possibility of a single outlier in a population. Data are expressed as mean ±SEM and considered statistically significant if *p* values were < 0.05.

## Accession codes

Original microarray data have been assigned NCBI Gene Expression Omnibus database (GEO) accession no. GSE57738.

| Reagent or Resource | Source and Identifier |
| --- | --- |
| *Antibodies* | |
| Anti-BrdU, FITC, clone PRB-1 | (BioLegend Cat# 515406, RRID:AB_2566333) |
| Anti-human GzB, A647, clone GB11 | (BioLegend Cat# 515403, RRID:AB_2114575) |
| Anti-human GzB, FITC, clone GB11 | (BioLegend Cat# 103204, RRID:AB_312989) |

*Continued on next page*

*Continued*

| Reagent or Resource | Source and Identifier |
| --- | --- |
| Anti-mouse B220, biotinylated, clone RA3-6B2 | (BioLegend Cat# 107005, RRID:AB_313300) |
| Anti-mouse CCR5 PE, clone HMCCR5 | (BioLegend Cat# 121606, RRID:AB_572007) |
| Anti-mouse CD11b, biotinylated, clone M1/70 | (BioLegend Cat# 101204, RRID:AB_312787) |
| Anti-mouse CD11c, PE, clone N418 | (BioLegend Cat# 117308, RRID:AB_313777) |
| Anti-mouse CD16/CD32, clone 93 | (BioLegend Cat# 101302, RRID:AB_312801) |
| Anti-mouse CD3, FITC, clone 145-2C11 | (BioLegend Cat# 100306, RRID:AB_312671) |
| Anti-mouse CD4, BV605, clone GK 1.5 | (BioLegend Cat# 100451, RRID:AB_2564591) |
| Anti-mouse CD4, PE, clone GK 1.5 | (BioLegend Cat# 100408, RRID:AB_312693) |
| Anti-mouse CD44, FITC, clone IM7 | (BD Biosciences Cat# 553133, RRID: AB_2076224) |
| Anti-mouse CD45.1, PB, clone A20 | (BioLegend Cat# 110722, RRID:AB_492866) |
| Anti-mouse CD45.2, APC-Cy7, clone 104 | (BioLegend Cat# 109824, RRID:AB_830789) |
| Anti-mouse CD62L, PE, clone MEL-14 | (BioLegend Cat# 104407, RRID:AB_313094) |
| Anti-mouse CD8a, biotinylated, clone 53-6.7 | (BD Biosciences Cat# 553029, RRID:AB_394567) |
| Anti-mouse CD8a, PE, clone 53-6.7 | (BioLegend Cat# 100708, RRID:AB_312747) |
| Anti-mouse CD8a, PE-Cy7, clone 53-6.7 | (BioLegend Cat# 100722, RRID:AB_312761) |
| Anti-mouse IFNg, PE-Cy7, clone XMG1.2 | (BioLegend Cat# 505826, RRID:AB_2295770) |
| Anti-mouse IFNg, APC, clone XMG1.2 | (BioLegend Cat# 505810, RRID:AB_315404) |
| Anti-mouse IL-1R1, PE, clone JAMA147 | (BioLegend Cat# 113505, RRID:AB_2125036) |
| Anti-mouse IL-18Ra, APC, clone BG/IL18RA | (BioLegend Cat# 132903, RRID:AB_2123952) |
| Anti-mouse NK1.1, biotinylated, clone PK136 | (BD Biosciences Cat# 553163, RRID:AB_394675) |
| Anti-mouse Tbet, PE, clone 4B10 | (BioLegend Cat# 644810, RRID:AB_2200542) |
| Anti-mouse TCRb, BV421, clone H57-597 | (BioLegend Cat# 109229, RRID:AB_10933263) |
| Anti-mouse TCRgd, biotinylated, clone GL3 | (BD Biosciences Cat# 553176, RRID:AB_394687) |
| Streptavidin, PE | (BioLegend Cat# 133505, RRID:AB_1626229) |
| Streptavidin, PE-Cy7 | (BD Biosciences Cat# 554061, RRID:AB_10053328) |
| *Chemicals, Peptides* | |
| 2-mercaptoethanol | SIGMA-Aldrich Cat# M6250 |
| 5-Bromo-2'-deoxyuridine (BrdU) | SIGMA-Aldrich Cat# B8434 |
| Collagenase Tipo IV | SIGMA-Aldrich Cat# C5138-16 |
| DNAse I | SIGMA-Aldrich Cat# DN25-16 |
| Guanidine thiocyanate | SIGMA-Aldrich Cat# 69277-2506 |
| Monensin | SIGMA-Aldrich Cat# 22373-78-0 |
| Paraformaldehyde | SIGMA-Aldrich Cat# 30525-89-4 |
| Saponin | SIGMA-Aldrich Cat# 54521 |
| Sodium azide | SIGMA-Aldrich Cat# 52002-1006 |
| Trizol | Ambion-LifeTechnologies Cat# 15596018 |
| Tskb20 peptide | Genscript |
| Tween 20 | Biorad Cat# 170-6531 |
| RPMI media | GIBCO Cat# 22400-071 |
| Fetal calf serum | GIBCO Cat# 12657-029 |
| PBS - Phosphate-Buffered Saline | GIBCO Cat# 10010031 |
| *Critical commercial assays* | |
| SYBR™ Green PCR Master Mix | Applied Biosystems Cat# 4309155 |

*Continued on next page*

*Continued*

| Reagent or Resource | Source and Identifier |
|---|---|
| IL-1beta/IL-1F2 DuoSet ELISA kit | R&D Systems Cat# DY401 |
| IL-18/IL-1F4 ELISA kit | R&D Systems Cat# 7625 |
| Dynabeads Biotin Binder | Invitrogen Cat# 11047 |
| Illumina MouseRef-8 v2.0 Expression BeadChips | Illumina Cat# BD-202-0202 |
| *Experimental models: Organisms/Strains* | |
| Mouse: C57BL/6 | RRID:IMSR_JAX:000664 |
| Mouse: B6.SJL | RRID:IMSR_JAX:002014 |
| Mouse: B6 x B6.SJL (F1) | animal house at UFF, Niteroi, BR |
| Mouse: Myd88-/- | RRID:MGI:5447806 |
| Mouse: IL18r1 | RRID:IMSR_JAX:004131 |
| Mouse: IL1r1-/- | RRID:IMSR_JAX:003245 |
| Mouse: Rag2-/- | RRID:IMSR_JAX:008449 |
| Trypanosoma cruzi | NCBI Taxon ID:5693 |
| *Oligonucleotides* | |
| Primers for qPCR | IDT - Integrated DNA Technologies |
| *Deposited data* | |
| Microarray | This paper GEO: GSE57738 |
| *Software* | |
| Summit V4.3 software | Dako Colorado, Inc |
| GeneSpring software GX 11.0 | Agilent Technologies |
| GraphPad Prism version 5 for Windows | GraphPad Software |
| GEO2R | NCBI NIH http://ncbi.nlm.nih.gov/geo/geo2r/ |

# Acknowledgements

This work was supported by Fundação Carlos Chagas Filho de Amparo à Pesquisa do Estado do Rio de Janeiro (FAPERJ), Conselho Nacional de Pesquisas (CNPq) and by The National Institute of Science and Technology for Vaccines (INCTV), Conselho Nacional de Desenvolvimento Científico e Tecnologico - CNPq (573547/2008-4). AG and FBC received PhD fellowships from CNPq. AG and FBC received post-doc fellowship from CSF program, CNPq. JFG and CDB received MSc fellowships from FAPERJ and CNPq, respectively. AN, ACO and MB received PQ fellowship from CNPq and AN and MB received CNE fellowship from FAPERJ.

The funders had no role in study design, data collection and analysis, decision to publish, or preparation of the manuscript.

# Additional information

## Funding

| Funder | Grant reference number | Author |
|---|---|---|
| Conselho Nacional de Desenvolvimento Científico e Tecnológico | 402932/2012-9 | Maria Bellio |
| Fundação Carlos Chagas Filho de Amparo à Pesquisa do Estado do Rio de Janeiro | 103.078/2011 | Maria Bellio |

| Fundação Carlos Chagas Filho de Amparo à Pesquisa do Estado do Rio de Janeiro | 110.168/2013 | Maria Bellio |
| Conselho Nacional de Desenvolvimento Científico e Tecnológico | 307557/2010-3 | Maria Bellio |

The funders had no role in study design, data collection and interpretation, or the decision to submit the work for publication.

## Author contributions

Ana-Carolina Oliveira, Data curation, Formal analysis, Investigation, Methodology, Writing—review and editing; João Francisco Gomes-Neto, Data curation, Formal analysis, Validation, Investigation, Methodology; Carlos-Henrique Dantas Barbosa, Alessandra Granato, Investigation, Methodology; Bernardo S Reis, Data curation, Formal analysis, Investigation, Methodology; Bruno Maia Santos, Methodology; Rita Fucs, Resources; Fábio B Canto, Investigation, Methodology, Writing—review and editing; Helder I Nakaya, Data curation, Software, Formal analysis, Validation, Visualization; Alberto Nóbrega, Formal analysis, Funding acquisition, Validation, Investigation, Writing—review and editing; Maria Bellio, Conceptualization, Data curation, Formal analysis, Supervision, Funding acquisition, Validation, Investigation, Visualization, Writing—original draft, Project administration, Writing—review and editing

## Author ORCIDs

Ana-Carolina Oliveira http://orcid.org/0000-0002-0036-3720
Maria Bellio http://orcid.org/0000-0002-3360-2740

## Ethics

Animal experimentation: Experiments were conducted in strict accordance with guidelines of the Animal Care and Use Committee of the Federal University of Rio de Janeiro (Comitê de Ética do Centro de Ciências da Saúde CEUA-CCS/UFRJ). Procedures and animal protocols were approved by CEUA-CCS/UFRJ license n.: IMPPG022. Every effort was made to minimize suffering.

## Decision letter and Author response

Decision letter https://doi.org/10.7554/eLife.30883.024
Author response https://doi.org/10.7554/eLife.30883.025

# Additional files

## Supplementary files

• Transparent reporting form
DOI: https://doi.org/10.7554/eLife.30883.021

## Major datasets

The following dataset was generated:

| Author(s) | Year | Dataset title | Dataset URL | Database, license, and accessibility information |
|---|---|---|---|---|
| Oliveira AC, Gomes-Neto JF, Barbosa CHD, Granato A, Reis BS, Canto FB, Santos BM, Fucs R, Nakaya HI, Nóbrega A and Bellio M | 2017 | T cell-intrinsic MyD88 signaling in cognate immune response to intracellular parasite infection: crucial role for IL-18R | https://www.ncbi.nlm.nih.gov/geo/query/acc.cgi?acc=GSE57738 | Publicly available at NCBI Gene Expression Omnibus (accession no. GSE57738) |

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
