## [Decision Letter]

[Editors’ note: a previous version of this study was rejected after peer review, but the authors submitted for reconsideration. The first decision letter after peer review is shown below.]

Thank you for submitting your work entitled "T cell-intrinsic MyD88 signaling in cognate immune response to intracellular parasite infection: crucial role for IL-18R" for consideration by *eLife*. Your article has been reviewed by three peer reviewers, one of whom is a member of our Board of Reviewing Editors and the evaluation has been overseen by a Senior Editor. The reviewers have opted to remain anonymous.

Our decision has been reached after consultation between the reviewers. Based on these discussions and the individual reviews below, we regret to inform you that your work will not be considered further for publication in *eLife* at this time.

The area of research surrounding the need for MyD88 signaling in mediating protective immunity against infections is quite extensive. You, however, have embarked on a study of a somewhat contradictory aspect of MyD88 signaling by investigating the contribution of cytokine receptors, namely IL-1R and IL-18R, which together with MyD88 may be necessary to maintain Th1 responses to prevent infections. Thus, the observation that signaling by MyD88 and IL-18R in CD4 T cells from mice infected with *T. cruzi* might potentially by novel. However, on the basis of strength of the evidence that you present here, the conclusions in regard to the predominant role of IL-18R signaling in T cells following *T. cruzi* infection are not fully supported. Although your experimental approaches utilizing various bone marrow mixed chimeras combined with the various knockout mice are quite impressive, the experiments also lack robustness.

Unedited reviews from all three reviewers are included. Because of the potential novelty of your findings, it is strongly suggested that you perform all the suggested key experiments and respond to all other comments in a manner that would be acceptable by *eLife* and you are welcome to consider submitting your manuscript as a new submission to *eLife*.

Reviewer #1:

There is a substantial body of evidence suggesting the need for MyD88 signaling to mediate protective responses against infections. These authors are guided by several contradictory observations concerning the contribution of cytokine receptors, namely IL-1R and IL-18R, which together with MyD88 may be necessary to maintain Th1 responses necessary to prevent infections. They have embarked on a set of rather ambitious experiments, utilizing bone marrow mixed chimeras and *MyD88^-/-^, IL1R^-/-^* and *IL18R^-/-^* mice to test the hypothesis that the absence of MyD88 signaling involving IL-1R family in CD4 T cells might be important factor contributing to deficient Th1 cells in *MyD88^-/-^* mice. According to the evidence that they present, it is IL-18R/MyD88 signaling in CD4 T cells that is needed for a proper Th1 (IFN-γ) response, activation of memory genes, and protection from apoptosis in Trypanosoma cruzi infected mice. The observation that signaling by MyD88 and IL-18R in CD4 T cells from mice infected with *T. cruzi* might indeed by novel.

However, I find this rather ambitious and extensive study to be lacking rigorous experimental approach. The *Rag2^-/-^* results are quite important in support of their hypothesis, yet they are relegated to the supplemental section. Moreover, gating strategies need to be shown for the cells isolated from the Rag^-/-^ recipients. Apart from expressing the results as% , the authors need to include cell numbers. This is true for all other results where the authors have chosen to express cellular responses as percentages. This is simply insufficient. In relation to the cell numbers, what proportion of CD4 T cells are specific for *T. cruzi* following infection? Is it possible just to sort all of the CD45.1 and then CD4^+^ T cells and assume that all of them are Tc specific?

Another serious concern is the number of mice used per experiment or the number of times a given experiment was performed. In general I find it difficult to trust statistics performed on responses from 3 or 4 mice. Also, what do the authors mean that a given experiment was performed either 3 or 4 times?

What are the percentages of CD45.2 vs. CD45.1 in the plot 1. Figure 3?

It also appears that results shown in Figure 4, plot A are the same as Figure 3. Is that true? The authors state in subsection “Lack of T cell-intrinsic IL-18R or MyD88 signaling leads to lower frequencies of Th1 cells but does not affect CD8^+^ CTLs” these results were from a different set of chimeras.

Is it possible that a transfer of a higher numbers of sorted CD4 T cells (Figure 6) would have been more effective in *MyD88^-/-^* recipients?

In addition, the writing itself is missing many details that detracts from my general enthusiasm about this body of work. I found the paper very difficult to read as I had to constantly move from the Results section to the Figure legends to find some necessary bit of information, which often was either contradictory or missing (Legends for Figure 1 and Figure 1—figure supplement 1). The results concerning CD11c DC appear (Figure 1—figure supplement 1) out of nowhere and one is not clear if these data are derived from LN or spleens and at which time point after infection. The same is true for using the Kb-restricted TSKB20 peptide. I would think that a preface for including CCR5 in the analysis is missing hence one is not clear as to why it was even measured.

Reviewer #2:

In this study, Oliveira, Gomes-Neto and colleague demonstrate that IL18R signaling has a T cell-intrinsic role in inducing Th1 responses against *T. cruzi*. I think this is an important and significant study for two reasons

a) It solves a controversial issue, ie. The role of MyD88, TLR and IL-1 in the induction of Th1 responses in *T. cruzi* infection.

b) It advances our understanding of CD4^+^ T cell responses in pathogenic infections.

The paper is novel, well organized and performed, clearly written

I have few questions that the authors may address:

1) Why is IL18 crucial for expansion of T cells but not IFN-γ production on a per cell basis? The authors do not discuss this interesting observation.

2) How much IL18 and how much IL1 are produced during *T. cruzi* infection. Can the authors measure serum levels of IL18 and IL1 at different timepoints?

3) What are the major sources of IL18 in *T. cruzi* infection? Antigen presenting cells or stromal cells?

4) What are the stimuli that induce I18 production? Which stimuli induce IL18 during *T. cruzi*?

Reviewer #3:

Overall, this work seems to lack significant conceptual advance to be of interest to the broader readership of *eLife*. Furthermore, it is unclear whether the presented data completely support the conclusions the authors have drawn in regard to the predominant role of IL-18R signaling in T cells following *T. cruzi* infection. Specifically:

1) Staining for IL-18R and IL-1R on T cells from infected mice should be performed, potentially with an assessment of IL1-R family receptor expression on parasite antigen-specific T cells.

2) The authors suggest a crucial role for IL-18R, however, the presented data in Figure 4 of T cell phenotypes suggest an additional role for IL-1R on CD4^+^ T cells.

3) Although a statistically significant contribution of IL-1R to the frequency of CD44^hi^ CD4^+^ and BrdU^+^ cells is seen (Figure 4), IFN-γ expression seems unaffected in the absence of IL-1R (Figure 3). It is unclear why WT cells in the WT:IL1-R mixed BM chimeras in Figure 3 are not making as much IFN-γ as the WT cells within the other mixed chimeras (WT:Myd88; WT:IL-18R) – this may suggest a greater, unexplored role for non-T cells. These additional signals may be absent when half of the non-T cells also lack expression of MyD88 or IL-18R.

4) The in vivo survival data suggest that IL-18R does not account for all of the MyD88-dependent responses, reflecting previous studies that TLR signaling and other IL-1R family members may play a role in T cell expansion, survival, activation, and/or IFN-γ production.

5) Parasite clearance seem to be largely unaffected in the absence of IL-18R signaling, both by time course parasitemia (Figure 5) and parasite load in the myocardium at day 14 p.i. (Figure 5). These data would suggest that, following pathogen clearance, other MyD88-dependent mechanisms are accounting for the observed survival differences. It would be important to know if T cells are mediating these effects since the authors also suggest MyD88-dependent signaling in the tissue may be largely responsible for parasite control.

---

## [Author Response]

[Editors’ note: the author responses to the first round of peer review follow.]

Reviewer #1:[…] The Rag2^-/-^ results are quite important in support of their hypothesis, yet they are relegated to the supplemental section.

Mixed bone-marrow chimeras have been classically employed in order to test the in vivorole of molecules intrinsic to cells of hematopoietic origin. In these studies, as in ours, a 1:1 mixture of wild type and knockout/mutant derived BM reconstitute irradiate recipients. Alternatively, when the cell type being investigated is a T lymphocyte, *Rag^-/-^*mice can be reconstituted with a mixture of total spleen, or purified mature T cells, derived from different donors, without the need of host irradiation. However, in this case, the B cell compartment is not fully reconstituted and antigen-presenting cells (APCs) of donor origin are minimally present (if at all) in the *Rag^-/-^*host. In our manuscript, the focus of study is the CD4^+^ T cell population in a model of infection with *T. cruzi*. Survival to infection with *T. cruzi* also depends on the presence of a B cell response (Cardillo et al., 2007). Consequently, due to the lack of B cells, the T-cell reconstituted *Rag^-/-^*mice are quite susceptible to the infection, even when reconstituted with 100% WT T cells, dying early after infection. Therefore, we concluded that, for this infection model, mixed-BM chimeric mice were more appropriate to test the role of T-cell intrinsic expression of the IL-1R/MyD88 and IL-18R/MyD88 signaling pathways for Th1 differentiation and expansion. Nevertheless, we did control experiments using 1:1 WT:*MyD88^-/-^* reconstituted *Rag2^-/-^* mice to rule out effects of body irradiation and to compare WT and *MyD88^-/-^* CD4^+^ T cells in a system where APCs are basically 100% WT, since the possibility (although remote) existed that T cell activation might be affected in irradiated BM-reconstituted hosts, in which part of the APCs lack expression of MyD88. As already shown in Figure 1—figure supplement 1 in the first version of the manuscript, the data obtained in reconstituted *Rag2^-/-^* mice confirmed the results obtained in the irradiated mixed BM chimeras. Since the results we’ve obtained in the two models were essentially the same, we originally thought it was better to show one representative experiment with reconstituted *Rag2^-/-^* mice in Figure 1—figure supplement 1. But following reviewer #1 request, we have moved these results to novel Figure 2, now including gating strategies and the percentage and absolute numbers of WT and *MyD88^-/-^* CD4^+^ T cells producing IFN-γ in *Rag2^-/-^* mice. We have also created Figure 2—figure supplement 1 to display gating strategies and the percentage and absolute numbers of WT and *MyD88^-/-^* CD8^+^ T cells producing IFN-γ in reconstituted *Rag2^-/-^* mice. Please note that, in order to increase the levels of B cells, *Rag2^-/-^* mice reconstituted 6 week earlier with a 1:1 mixture of WT and *MyD88^-/-^* splenocytes, received a second injection of 1:1 mixture of WT and *MyD88^-/-^* splenocytes 6 hrs before being infected, as now described in the Materials and methods section.

Moreover, gating strategies need to be shown for the cells isolated from the Rag^-/-^ recipients.

Gating strategies for cells from the reconstituted *Rag2^-/-^* mice are now shown in Figure 2 and Figure 2—figure supplement 1.

Apart from expressing the results as% , the authors need to include cell numbers. This is true for all other results where the authors have chosen to express cellular responses as percentages. This is simply insufficient.

We agree with reviewer #1 that besides percentages, absolute number of cells may be relevant for the interpretation of the results. These data were now added to the revised version of the manuscript (see Figure 2, Figure 2—figure supplement 1, Figure 5, Figure 6—figure supplement 1 to 5 and Figure 7), as requested.

In relation to the cell numbers, what proportion of CD4 T cells are specific for *T. cruzi* following infection? Is it possible just to sort all of the CD45.1 and then CD4^+^ T cells and assume that all of them are Tc specific?

We have some difficulty in understanding which is exactly the point raised by the reviewer here, as we do not claim at any moment in the manuscript that the totality of sorted CD4^+^ T cells in experiment shown in Figure 2 (now Figure 3) is *T. cruzi*-specific. In the past, massive T cell responses to *T. cruzi,* as well as to other pathogens, such as certain viruses, were interpreted as mostly polyclonal and nonspecific. However, with the advent of MHC-tetramer technology and other pieces of evidence, it was shown that a higher percentage of CD8^+^ T cells activated during acute viral infection was in fact virus-specific, than had previously been estimated. In fact, up to 50% of CD8^+^ T cells responding during the peak of viral infection can be specific for a single epitope (Murali-Krishna et al., 1998). Immunodominant *T. cruzi* CD8 epitopes, as well as a large number of specific subdominant CD8 epitopes, have also been described. These studies have revealed that selected members of the *trans*-sialidase gene family were natural targets for *T. cruzi* infection-induced CD8^+^ T cells, which can represent >30% of the entire CD8 compartment at the peak of the response in mice (reviewed in Tarleton 2015). As trypanosomatids have an extensively larger genome than viruses, one may expect that a larger number of pathogen-derived epitopes is presented to T cells. However, to the present, the exact proportion or numbers of specific CD4^+^ T cells against *T. cruzi* have not been fully evaluated. Moreover, defined immunodominant I-Ab-restricted CD4 epitopes derived from the Y-strain have not been described yet. Nevertheless, different studies have demonstrated that a significant proportion of CD4^+^ T cells produces IFN-γ in response to parasite Ags in infected humans and mice and that these responses are not a generalized non-specific activation of cells during infection with *T. cruzi* (Martin & Tarleton 2005). In our present manuscript, as well as in previous works of our group and from others (Oliveira et al., 2010), 4 to 12% of the CD4^+^ T cells from the spleen of B6 infected mice secreted IFN-γ in an Ag-dependent way. Moreover, as shown in Figure 6, we found that the percentage of cells expressing high levels of the activation/memory marker CD44, among CD4^+^ T lymphocytes in infected chimeras at day 14 pi, varies from 55% (among *MyD88^-/-^* CD4^+^) to 85% (among WT CD4^+^). Absolute numbers of CD44high CD4^+^ T cells are now shown in Figure 6—figure supplement 5, also indicating a huge increase in the number of activated/memory cells. Moreover, Figure 4 now shows that, in infected mice, most CD44hiCD4^+^ T cells do not express CD62L, which is the phenotype of effector and effector memory (TEM) cells. On the other hand, around 10% of total CD4^+^ T cells display the CD44^hi^CD62L^+^ TCM (central memory) phenotype at day 14 pi. Previous works have also demonstrated that high numbers of CD4^+^ TEM cells are generated in response to *T. cruzi* (Martin & Tarleton 2005). Therefore, activated effector and both TEM and TCM memory CD4^+^ T cells are highly represented in our sorting experiment, mainly among WT cells, and the results obtained in the microarray essay of sorted CD4^+^ T cells simply reflect this fact.

Another serious concern is the number of mice used per experiment or the number of times a given experiment was performed. In general I find it difficult to trust statistics performed on responses from 3 or 4 mice. Also, what do the authors mean that a given experiment was performed either 3 or 4 times?

Data illustrated in Figure 3 and Figure 4 (now Figure 5 and Figure 6), were obtained with 3 non-infected chimeric controls and 3 infected chimeric mice from each of the 4 groups of mixed-BM chimeras (WT:WT, WT:*MyD88^-/-^*, WT:*Il1r1^-/-^* and WT:*Il18r1^-/-^*^),^ which were simultaneously assayed, in a total of 24 mice in each experiment. Student´s t-test was used for comparing cytokine and markers expression between WT and KO T cells within each group of chimeric mice. The small sample size, n=3, only allows statistical detection at p<0,05 if the effect is large and exhibit low variance, thus supporting our interpretation of the results. Moreover, group size n=3 is routinely used when comparing cytokine intracellular staining in lymphocytes by cytometry (examples are: Yang et al., 2016; Brennan et al., 2016, Figure 5; Ho et al., Figure 4) and, thus, our study procedure does not differ from the one adopted in papers published in renowned journals. However, in order to convince reviewer #1 of the consistency of our data and further strengthen the statistics, we have now combined data from two independent experiments, and a total of 6 to 8 mice per group are shown in Figure 5, Figure 6 and Figure 6—figure supplement 1 to 5. (Only in WT:WT control chimeras, the group size is n=3).

The meaning of a given experiment having been performed 3 to 4 times is the following: different parameters were tested repeatedly in independent experiments: for example, BrdU incorporation was performed in 3 independent experiments, while intracellular IFN-γ staining was performed in 4 independent experiments. In each independent experiment, new chimeric mice were obtained by irradiating WT recipients that were then divided into 4 groups, each one being reconstituted with one of the different 1:1 mixtures of WT and KO BM and were infected 6 to 8 weeks later. In some experiments, only WT:*MyD88^-/-^* chimeras were constructed. In each independent experiment, 3 controls and 3 to 4 infected mice of each set of chimeras had the expression of several markers analyzed by flow cytometry.

What are the percentages of CD45.2 vs. CD45.1 in the plot 1. Figure 3?

Percentages of CD4^+^ CD45.2^+^ and of CD4^+^ CD45.1 in the plot 1 of Figure 3 (now Figure 5) are 36.53% and 38.35%, respectively. Percentages were now added to the Figure plot.

It also appears that results shown in Figure 4, plot A are the same as Figure 3. Is that true? The authors state in subsection “Lack of T cell-intrinsic IL-18R or MyD88 signaling leads to lower frequencies of Th1 cells but does not affect CD8^+^ CTLs” these results were from a different set of chimeras.

Yes, Figure 4—figure supplement 1: (now Figure 6—figure supplement 1), plot A are the same as Figure 3 (now Figure 5), since the stainings analyzed on both Figures are from the very same experiment. In order to avoid redundancy, however, we have now deleted this dot plot form Figure 6—figure supplement 1 and we apologize for this.

It seems there was a misunderstanding about text in subsection “Lack of T cell-intrinsic IL-18R or MyD88 signaling leads to lower frequencies of Th1 cells but does not affect CD8+ CTLs”: In fact, we stated that: “We then analyzed BrdU incorporation in vivo by CD4^+^ T cells in the different sets of infected chimeras”. In fact, we were referring to the 3 previous cited different groups of mixed BM chimeras: *MyD88^-/-^*:WT; *Il1r1^-/-^*:WT and *Il18r1^-/-^*:WT, which were already analyzed in the previous Figure 3 (now Figure 5) To make it clearer we have now changed it for: “in the same groups of mixed BM chimeras”.

Is it possible that a transfer of a higher numbers of sorted CD4 T cells (Figure 6) would have been more effective in MyD88^-/-^ recipients?

This is an interesting point. We have done this experiment, now injecting 4 x 10^6^ purified WT CD4^+^ T cells, that is, 3 times more cells than in the experiment shown in Figure 6, with the same results, as shown in Author response image 1. Therefore, a significantly increased number of transferred CD4 T cells was not more effective in protecting *MyD88^-/-^* recipients against mortality induced by infection.

This information was now added to the text of the revised manuscript, in subsection “*Il18r1^-/-^* mice are highly susceptible to infection with *T. cruzi*” as data not shown.

In addition, the writing itself is missing many details that detracts from my general enthusiasm about this body of work. I found the paper very difficult to read as I had to constantly move from the Results section to the Figure legends to find some necessary bit of information, which often was either contradictory or missing (Legends for Figure 1 and Figure 1—figure supplement 1). The results concerning CD11c DC appear (Figure 1—figure supplement 1) out of nowhere and one is not clear if these data are derived from LN or spleens and at which time point after infection.

We apologize if our text was not enough clear for reviewer #1 understanding and we have now made modifications to improve it. However, it seems reasonable to us that details about experiments should be displayed in the Figure legends and in the Materials and methods section. Otherwise, an over-detailed text would lack fluidity and result even more difficult to read. In the specific case concerning the dot plot showing the frequencies of CD45.1^+^ (WT) and CD45.2^+^ (*MyD88^-/-^*) CD11c^+^ cells (Figure 1—figure supplement 1), we referred to it in the main text (subsection “*MyD88^-/-^* Th1 cells attain lower frequencies in infected mixed BM chimeras.”) as follows: “Including residual recipient cells (CD45.1^+^CD45.2^+^), the mixed BM chimeras have more than 50% of WT DC in their LNs and spleen (Figure 1—figure supplement 1), which can be fully activated by TLR pathways during infection with *T. cruzi*.” Therefore, this result did not “appear out of nowhere”, but instead it is referred to in the main text and it constitutes an important control about the frequency of WT DCs in the mixed BM chimeras. Moreover, in the legend of Figure 1—figure supplement 1, we state that “(B) Representative frequencies of CD45.1^+^ (B6.SJL, WT) and CD45.2^+^ (*MyD88^-/-^*) cells gated on CD11c^hi^ spleen cells of non-infected mixed BM chimeric mice.” Therefore, it seems to us that the only missing information is the specification that equal results obtained in LNs were not shown. We have now removed the mention to LNs and apologize for this little forgetfulness. In the present version of the manuscript, we have also added that residual recipient (CD45.1^+^CD45.2^+^) cells are of WT origin, although this was already said in the description of mixed BM chimeras, few lines above: “irradiated WT B6 x B6.SJL F1 (CD45.1^+^CD45.2^+^), mice were reconstituted […]” (subsection “*MyD88^-/-^* Th1 cells attain lower frequencies in infected mixed BM chimeras.”).

The same is true for using the Kb-restricted TSKB20 peptide.

It is well known that effector T cells need to be further stimulated in vitro in the presence of monensin or brefeldin for cytokine intracellular staining (ICS) to succeed. In the *T. cruzi* model, while the pathogen-derived Ags present in the infected total spleen culture are sufficient for stimulating IFN-γ production by CD4^+^ T cells, the addition of specific antigenic peptides to the cultures is needed for the detection of IFN-γ in CD8^+^ T cells by ICS. In fact, when purified CD4^+^ T cells from infected mice were stimulated in vitro with non-infected APC, in the absence of any *T. cruzi* antigen, no IFN-γ is detected, as previously shown (Bafica et al., 2006 Figure 3). On the other hand, when total splenocytes from infected mice on day 14 pi are plated, infected macrophages and DCs are present in the cultures and are sufficient for the Ag-specific stimulation of CD4^+^ T cells. In a previous study, we have shown that adding *T. cruzi* amastigote protein extract to the cultures only marginally increases the release of IFN-γ by CD4^+^ T cells in total splenocyte cultures (Oliveira et al., 2010). As immunodominant CD4 epitopes of *T. cruzi* have not been reported, we have chosen to stimulate CD4^+^ T cells in cultures using infected total splenocytes. For CD8^+^ T cells, on the other hand, immunodominant epitopes are well characterized, therefore, we added the immunodominant TSKB20 peptide to the cultures in order to further increase the specific antigenic response of CD8^+^ T cells. To explain this, the following phrases were present in the first version of the manuscript (subsection “*MyD88^-/-^* Th1 cells attain lower frequencies in infected mixed BM chimeras.”): “In order to determine the frequency of Ag-specific IFN-γ-producing T cells in the different T cell subpopulations, total splenocytes from infected mixed BM chimeras were cultured in the presence of the Kb-restricted TSKB20 peptide, an immunodominant CD8 epitope (Oliveira et al., 2010). Infected APCs present in the spleen of infected mice are able to induce Ag-specific stimulation of CD4^+^ T cells in vitro without the need of adding extra *T. cruzi*-derived Ag in the in vitro cultures (Oliveira et al., 2010)”.

We believe this is sufficiently clear, but we will be glad to further clarify this point whether the reviewer believes it is still necessary.

I would think that a preface for including CCR5 in the analysis is missing hence one is not clear as to why it was even measured.

It is known that although chemokine receptor expression and differentiated Th phenotype are not strictly coordinate, some receptors, such as CXCR3 and CCR5, show a striking preferential expression on Th1 cells (Zhu & Paul, 2008; Bonecchi et al., 1998). We first referred to CCR5 in subsection “WT and *MyD88^-/-^* CD4^+^ T cells of infected mixed BM chimeras display different gene-expression programs.”, when introducing RNA microarray results: “Several Th1 related genes as IFN-γ, Ccr5 Ccl5, Ccl4 and Ccl3 were upregulated in WT CD4^+^ T cells compared to *MyD88^-/-^* CD4^+^ T cells”. Therefore, we have next analyzed, by flow cytometry, the expression of CCR5, the receptor for RANTES, macrophage-inflammatory protein 1α (MIP-1α) and MIP-1β on CD4^+^ T cells from mixed BM chimeras. We confirmed that a higher percentage and absolute number of CD4^+^ T cells expresses CCR5+ among WT, than among *MyD88^-/-^* or *IL18R^-/-^* CD4^+^ T cells (Figure 6 and Figure 6—figure supplement 2) and referred to this result as: “Th1- associated chemokine receptor CCR5 among WT, *MyD88^-/-^*, *Il1r1^-/-^* or *Il18r1^-/-^* CD4^+^ T lymphocytes […]”(subsection “Lack of T cell-intrinsic IL-18R or MyD88 signaling leads to lower frequencies of Th1 cells but does not affect CD8+ CTLs”).

This result is in accordance with our microarray data and, together with the other results of the manuscript, led us to the conclusion that a higher expansion of Th1 cells occurs among CD4^+^ T cells expressing the IL-18/Myd88 signaling pathway.

Reviewer #2:1) Why is IL18 crucial for expansion of T cells but not IFN-γ production on a per cell basis? The authors do not discuss this interesting observation

A low number of Th1 cells might be able to differentiate, proliferate and survive despite the lack of IL-18R/MyD88 signaling, possibly due to alternative signaling pathways, as IL-27 and/or others. Apparently, however, these alternative signaling pathways would not be as efficient as IL-18R/MyD88 for the expansion/survival of Th1 cells, at least during infection with *T. cruzi.* Nevertheless, it appears that the few IL-18R/MyD88-deficient Th1 cells, which are able to differentiate and survive, are also able of producing as much IFN-γ (induced by another pathway) as WT Th1, on a per cell basis. Since we have not experimentally explored this question in more detail, we refrained to speculate more on this observation in the present manuscript and this result was removed from the present version of the manuscript.

2) How much IL18 and how much IL1 are produced during T. cruzi infection. Can the authors measure serum levels of IL18 and IL1 at different timepoints?

We performed a kinetic study of the presence of IL-1β and IL-18 in the sera of infected B6 mice. These data are now included in the manuscript in Figure 4—figure supplement 1. As shown, a higher level of IL-1β was detected on day 13 pi, although an early peak of this cytokine was also detected at 12 h after infection. On the other hand, we could only detect IL-18 at later time points of infection (day 13 pi). This result is in accordance with the kinetics of IL-18 detection in the supernatants of spleen cultures of infected C3H mouse strain, previously published (Antunez et al., 2001). Detection of IL-18 in the serum at 13 dpi was also paralleled by IL-18 mRNA detection in the spleen, previously reported by others (Muller et al., 2001). Peaks of IL-1β and IL-18 at day 13 pi were followed by increased percentages of Th1 cells in the spleen of infected mice (Figure 4—figure supplement 1). It is important to note, however, that the biology of IL-1-family cytokines is rather complex: soluble receptors, as well as IL-binding protein, such as IL-1R2 and IL-18BP, might be present in the sera and inhibit IL-1 and IL-18 function, respectively, as well as their detection (Arend et al., 1994; Nakanishi et al., 2001). Moreover, IL-1β signaling can be outcompeted by the IL-1Ra soluble molecule, which is an antagonist that binds IL-1R1 with an affinity higher than that of IL-1β but fails to recruit the signaling IL-1RAcP chain.

3) What are the major sources of IL18 in *T. cruzi* infection? Antigen presenting cells or stromal cells?

This is an interesting question. IL-18 mRNA is expressed in a wide range of cells including Kupffer cells, macrophages, T cells, B cells, dendritic cells, osteoblasts, keratinocytes, astrocytes, and microglias (reviewed in Nakanishi et al., 2001). As *T. cruzi* is able to infect different organs as liver and brain, among others, it is possible that different cell types, at different locations, are producing IL-18 following infection. However, few studies have fully investigated this point at the present, to our knowledge. IL-18 transcripts were detected in total spleen cells from infected B6 mice, increasing until day 14 post-infection (Muller et al., 2001). Also, IL-18 was detected in the supernatants of splenocyte cultures from infected mice (Antunez & Cardoni, 2001). Moreover, it has been shown, by in situ hybridization, that the number of cells expressing IL-18 mRNA in the spleen increases during infection with *T. cruzi* and most of the positive cells were found in the white pulp, especially in the PALS (Meyer Zum Buschenfelde et al., 1997). Our preliminary data have shown that IL-18 could also be detected in the supernatant (SN) of myocardium cultures obtained from infected mice, although at lower levels than in SN of spleen cultures (data not shown).

4) What are the stimuli that induce I18 production? Which stimuli induce IL18 during *T cruzii*?

IL-18 gene expression may be upregulated by direct stimulation with microbe products such as LPS, or by cytokines such as type I and type II IFNs (reviewed in Nakanishi et al., 2001). Moreover, pro‐IL‐18 needs processing by caspase 1 to turn into functional IL-18. Thus, IL-18 production also requires inflammasome activation, or an alternative processing pathway (reviewed in Garlanda et al., 2013). Both TLR signaling pathways and the NALP3 inflammasome activation have been shown during infection with *T. cruzi* (Rodrigues et al., 2012; Silva et al., 2013). Specifically, glycosylphosphatidylinositol anchors of mucin-like glycoproteins (GPI-mucins), expressed in the surface of the trypomastigote stage of the parasite, have been shown to activate the TLR2 signaling pathway and to induce IL-18 transcripts (Ferreira et al., 2002; Campos et al., 2001). Moreover, one may expect that other described *T. cruzi*-derived PAMPs (as GIPL or CpG) together with danger signals released by tissue damage caused by the parasite and/or the inflammatory response against it, would also contribute to IL-18 production during infection.

Reviewer #3:Overall, this work seems to lack significant conceptual advance to be of interest to the broader readership of eLife.

The results described in our manuscript bring important advances for understanding the role of T cell-intrinsic IL-18/MyD88 signaling on the development of a robust Th1 response to infection. IL-18 has been known for a long time to synergize with IL-12 for Th1 cell differentiation in vitro, but its requirement for Th1 response in vivois controversial, possibly because of compensatory mechanisms present in *Il18* gene-deficient mice, as elevated IL-12 production. Also, during infection, IL-18 can play an indirect role on Th1 cells, due to its effect of inducing IFN-γ in NK cells, which in turn also play a key role in Th1 differentiation. On the other hand, some studies have shown that T-cell intrinsic MyD88-mediated signaling severely impact the immune response. However, the TIR domain-containing receptor upstream of MyD88 acting on CD4^+^ T cells was either not investigated or not identified and, therefore, this topic remains speculative and needs further clarification. By excluding other cell types, as NK cells and the IL-1R receptor-signaling pathway in this process, our work delimitates a mechanistic framework and brings a conceptual advance to the field. To the best of our knowledge, our study is the first to formally and unequivocally demonstrate the crucial role of T-cell intrinsic IL-18R/MyD88 pathway for sustaining the Th1 response in a model of in vivoinfection, showing that IL-18R is a crucial MyD88-dependent receptor intrinsic to T cells necessary for Th1 expansion in vivo. Therefore, our manuscript answers a controversial open question of broad interest and, in our opinion, is suitable for publication in *eLife*.

Furthermore, it is unclear whether the presented data completely support the conclusions the authors have drawn in regard to the predominant role of IL-18R signaling in T cells following *T. cruzi* infection. Specifically:1) Staining for IL-18R and IL-1R on T cells from infected mice should be performed, potentially with an assessment of IL1-R family receptor expression on parasite antigen-specific T cells.

We have now included Figure 4, showing the expression of IL-18R and IL-1R on CD4^+^ T cells of the spleen of infected mice, by flow cytometry, as requested by the reviewer #3. As mentioned above (in answer to reviewer #1), there is no *T. cruzi* Y strain-specific immunodominant I-Ab-restricted peptide described in the literature. Therefore, it was not possible do perform tetramer-Ag staining. To circumvent this, we stained spleen cells with anti-CD4, anti-CD44, anti-CD62L and anti-T-bet (in addition to anti-IL18R and anti-IL-1R mAbs), which allow us to distinguish effector Th1 and memory CD4^+^ T cells. Our results show that while IL-18R is expressed by the majority of CD44^hi^CD62L^+^ (TCM) and CD44^h^iCD62L^-^ CD4^+^ (Teff and TEM) T cells, mainly among T-bet+ cells, IL-1R is expressed marginally only by CD44^hi^CD62L^+^CD4^+^ T cells (TCM). Importantly, no increase on the expression of IL-1R could be detected on any subpopulation of CD4^+^ T cells from the spleen of infected mice compared to non-infected controls. Therefore, our new added data further support our previous conclusions on the major role of IL-18R on the Th1 response to infection.

2) The authors suggest a crucial role for IL-18R, however, the presented data in Figure 4 of T cell phenotypes suggest an additional role for IL-1R on CD4^+^ T cells.

The only significant difference found between WT and *IL-1R^-/-^* T cells from WT:*IL-1R^-/-^*mixed BM chimeras, shown in Figure 4 (now Figure 6), concerned the percentages of CD44^hi^CD4^+^ T cells in the infected spleen. Therefore, we did not exclude a minor role for IL-1R and this was mentioned in our Discussion section. However, our data clearly excluded a role for IL-1R in the Th1 response. The new results now included in the revised manuscript on Figure 4, also show that IL-1R can only be detected on CD4^+^ T cells with central memory (TCM) CD44^hi^CD62L^+^ phenotype, both in non-infected and infected mice, while the expression of IL-18R increases in both effector and memory CD4^+^ T cells, following infection. These results add to the data presented in the first version of the manuscript, favouring a scenario where IL18R, but not IL-1R, plays a major role in the MyD88-mediated signaling, which leads to a robust Th1 response against infection.

3) Although a statistically significant contribution of IL-1R to the frequency of CD44^hi^ CD4^+^ and BrdU^+^ cells is seen (Figure 4), IFN-γ expression seems unaffected in the absence of IL-1R (Figure 3). It is unclear why WT cells in the WT:IL1-R mixed BM chimeras in Figure 3 are not making as much IFN-γ as the WT cells within the other mixed chimeras (WT:Myd88; WT:IL-18R) – this may suggest a greater, unexplored role for non-T cells. These additional signals may be absent when half of the non-T cells also lack expression of MyD88 or IL-18R.

We think there is some misunderstanding here: we found no statistical difference for BrdU incorporation between WT and IL-1R^-/-^CD4^+^ T cells from WT: IL-1R^-/-^ mixed BM chimeras. Significant difference between WT and IL-1R^-/-^CD4^+^ T cells was found only regarding the percentage of CD44^hi^ cells, as answered in the previous question. Concerning the possibility “that additional signals may be absent when half of the non-T cells also lack expression of MyD88 or IL-18R”, we believe that the results obtained with the non-irradiated *Rag2^-/-^* reconstituted model showed on new Figure 2 (former Figure 1—figure supplement 1) discard this hypothesis, since in this case the vast majority, if not the totality, of APCs is WT concerning Myd88 and Il18r genes. The results on IFN-γ production obtained in experiments illustrated on Figure 3 (now Figure 5) can be explained as follows: please note that, in the case of WT:WT chimera, both CD45.1^+^ and CD45.2^+^ T cells are able to produce IFN-γ and the sum of the frequencies of both IFN-γ -producing WT populations (≈ 6.3%) is equivalent to the total percentage of IFN-γ^+^ cells resulting from the sum of WT CD45.1+IFN-γ+ cells plus *MyD88^-/-^* CD45.2^+^IFN- γ^+^ in WT:*MyD88^-/-^* chimeras (≈7.4%). The same is true for the sum of total IFN-γ-producing cell frequency in WT:*IL18R^-/-^* (≈ 6.8%) or WT:*IL1R^-/-^* (≈ 6.0%) mixed chimeras. The same applies for absolute cell numbers (Figure 5). This indicates that there might be a control of the total number of CD4^+^ IFN-γ-producing cells in infected mice. Therefore, in the case of WT:WT and WT:*IL1R^-/-^* mixed chimeras, the total number of CD4^+^IFN-γ^+^cells is equally distributed between the two populations, while in WT:*MyD88^-/-^* and WT:*IL18R^-/-^* mixed BM chimeras, the maximum number of IFN-γ-producing cells is attained mainly within the CD45.1^+^ WT population, since the KO CD4^+^ T cells do not expand in these chimeras. Absolute numbers of cells in these experiments are now shown in Figure 5, further supporting our conclusion.

4) The in vivo survival data suggest that IL-18R does not account for all of the MyD88-dependent responses, reflecting previous studies that TLR signaling and other IL-1R family members may play a role in T cell expansion, survival, activation, and/or IFN-γ production.

Our present work demonstrates that IL-18R does give a crucial contribution to the MyD88-dependent responses intrinsic to CD4 T cells, which are necessary for a robust Th1 response. We did not claim, at any time, that the lack of IL-18R signaling accounts for the lack of all the MyD88-dependent responses in *MyD88^-/-^* mice, which include signaling by different receptors as TLRs and other IL-1R family members, in many different cell types besides T cells, such as monocytes, DCs, B cells, cardiomyocytes and others. However, the major focus of this work was exactly to study the role of IL-18R/MyD88 signaling intrinsic to T cells, during infection. Interestingly, we have also shown here, for the first time, that *IL18R^-/-^* mice are also very susceptible to infection with *T. cruzi* and can be rescued from death by receiving the adaptive transference of WT CD4^+^ T cells. This is in accordance with the results obtained in mixed BM chimeras on the importance of IL-18R signaling in Th1 cells and on the protective role of this cell subset against infection. We have discussed this issue in the Discussion section.

5) Parasite clearance seem to be largely unaffected in the absence of IL-18R signaling, both by time course parasitemia (Figure 5) and parasite load in the myocardium at day 14 p.i. (Figure 5). These data would suggest that, following pathogen clearance, other MyD88-dependent mechanisms are accounting for the observed survival differences. It would be important to know if T cells are mediating these effects since the authors also suggest MyD88-dependent signaling in the tissue may be largely responsible for parasite control.

We agree this is an important point. The role of CD4^+^ T cells in mediating parasite control is a fundamental question, and we have tested this point in the experiments illustrated on Figure 6 of the first version of the manuscript (now Figure 8). However, we believe there is some misunderstanding here, since two statements in the above comment by the reviewer do not apply. Firstly, results shown in Figure 5 undoubtedly demonstrate that: at day 9 pi, when the peak of parasites in the blood is attained, no statistical difference in parasitemia levels was found between Il18r1-/- and *MyD88^-/-^* mice, as stated in subsection “*Il18r1^-/-^* mice are highly susceptible to infection with *T. cruzi* “, while both strains of KO mice display significant higher parasitemia than WT mice (subsection “Lack of T cell-intrinsic IL-18R or MyD88 signaling leads to lower frequencies and numbers of Th1 cells but does not affect CD8^+^CTLs.”). This means that *Il18r1^-/-^* and *MyD88^-/-^* are equally susceptible in terms of parasitemia, while WT mice are more resistant.

Second, parasite load in the myocardium is also significantly higher in *Il18r1^-/-^* when compared to WT mice, as shown in Figure 5 (now Figure 7). Therefore, it is wrong to affirm that: “Parasite clearance seem to be largely unaffected in the absence of IL-18R signaling”, (both in the blood and in the myocardium), as stated by the reviewer. However, it is true that parasites are cleared from the blood of *Il18r1^-/-^* and *MyD88^-/-^* mice with essentially the same kinetics observed in WT mice. Also, *MyD88^-/-^* mice displayed higher levels of parasite in the tissue than *Il18r1^-/-^* mice. Since: (i) we have found the same ratios and absolute numbers of Th1 cells in the spleen of *Il18r1^-/-^* and *MyD88^-/-^* mice (Figure 5, Figure 6 and Figure 6—figure supplement 2) and (ii) we have previously shown that CD8-mediated specific cytotoxicity is totally preserved in *MyD88^-/-^* mice (Oliveira et al., 2010), we have raised the hypothesis on the putative importance of innate signaling through TLRs in the tissue for parasite control, which could explain the higher parasite levels observed in the myocardium of *MyD88^-/-^* mice, compared to Il18r1-/- mice (subsection “Il18r1-/- mice are highly susceptible to infection with *T. cruzi*”). In fact, it is known that TLR4 signaling and IFN-γ synergize for the induction of microbicidal activity in macrophages. Therefore, even though equivalent levels Th1-derived IFN- γ might be present in the infected tissues of Il18r1-/- and *MyD88^-/-^* mice, most TLR signaling is defective in *MyD88^-/-^* cells. As a consequence of this fact, higher parasite loads in *MyD88^-/-^* cells would result.

Thirdly, contrary to what is affirmed above by the reviewer, the pathogen is never cleared in *MyD88^-/-^* mice, which all die before day 15-18 pi, a time point when parasite load is still very high in the tissues. Therefore, it is not following parasite clearance that other MyD88-dependent mechanisms are accounting for the observed survival differences, as stated by reviewer #3, although it is possible that other MyD88-dependent mechanisms, not necessarily involved in parasite killing, may be responsible for survival. We hope these points are now clarified.

Concerning the role of T cells: in order to test the role of CD4^+^ T cells in parasitemia control and survival, we have performed the adoptive-transference experiments shown in Figure 6 (now Figure 8): while transferred WT CD4^+^ T cells can rescued *Il18r1^-/-^* mice from death, this treatment is not equally efficient in *MyD88^-/-^* infected mice, which mortality is significantly delayed, but not stopped. Parasitemia, on the other hand, is brought to WT levels in both *Il18r1^-/-^* and *MyD88^-/-^* mice receiving transferred WT CD4^+^ T cells. We also demonstrated that the capacity of lowering parasitemia depends on IFN-γ secretion by CD4^+^ T cells (Figure 6—figure supplement 1, now Figure 8—figure supplement 1). We interpreted these results as follows: since mortality following infection with *T. cruzi* is in some cases due to increased pro-inflammatory cytokine release (Holscher et al., 2000), it is possible that other MyD88-independent innate mechanisms, as the TLR4/TRIF pathway, active in *MyD88^-/-^* mice, might be responsible for this putative pro-inflammatory effect in this strain and responsible for death. Alternatively, but not mutually exclusive, tissue-tolerance mechanisms, possibly involving MyD88-dependent feedback control responses, might protect the host from immune- or pathogen-inflicted damage and would be impaired in *MyD88^-/-^* mice (Discussion section). Therefore, we believe the results shown in Figure 6 (now Figure 8) answer the question of reviewer #3 about the role of T cells in parasite control, although further work is necessary to fully uncover the reason for the failure of transferred WT CD4^+^ T cells to completely rescue *MyD88^-/-^* mice from death. However, this last point would require further extensive investigation and goes beyond the scope of the present manuscript.